# Dock-and-lock binding of SxIP ligands is required for stable and selective EB1 interactions

Teresa Almeida[1†], Eleanor Hargreaves[1†], Tobias Zech[2], Igor Barsukov[1*]

[1]Department of Biochemistry, Cell Signalling and Systems Biology, Institute of Systems, Molecular and Integrative Biology, University of Liverpool, Liverpool, United Kingdom; [2]Department of Molecular and Clinical Cancer Medicine, Institute of Systems, Molecular and Integrative Biology, University of Liverpool, Liverpool, United Kingdom

*For correspondence:
igb2@liverpool.ac.uk

[†]These authors contributed equally to this work

Competing interest: The authors declare that no competing interests exist.

## eLife Assessment

This study provides **important** insights into how the EBH domain of microtubule end-binding protein 1 (EB1) interacts with SxIP peptides derived from the MACF plus-end tracking protein. The revised manuscript includes **convincing** ITC and NMR experiments that clarify the role of flanking residues and address the influence of dimerization and cooperativity on binding. While some mechanistic aspects remain difficult to resolve experimentally, the data and analysis now more clearly justify the proposed "dock-and-lock" model and its interpretive value. This work will be of interest to structural biologists and biophysicists studying microtubule-associated protein interactions.

**Abstract** End binding protein 1 (EB1) is a key component of the signalling networks at microtubule plus ends. It contains an N-terminal microtubule-binding CH domain and a C-terminal EBH domain interacting with SxIP-containing sequences of other microtubule plus end tracking proteins (+TIPs). Using a series of SxIP-containing peptides derived from the microtubule-actin cross-linking factor (MACF), we demonstrate that the SxIP motif itself binds to EBH with low affinity, and that the full interaction requires contribution of residues following the SxIP motif. Based on the solution structure and dynamics of the EBH/MACF complex, we propose a two-step 'dock-and-lock' model for the EBH interaction with targets, where the SxIP motif initially binds to a partially formed EBH pocket. This subsequently induces folding of the unstructured C-terminus and transition to a stable complex. We dissect contributions into the binding and design MACF mutations of the post-SxIP region that enhance the affinity by two orders of magnitude, leading to a nanomolar interaction. We verify the enhanced recruitment of the mutated peptide to the dynamic plus ends of MTs in a live-cell experiment. Our model explains EB1's interaction with the SxIP-containing ligands and can be used to design small-molecule inhibitors blocking SxIP interaction with EB1.

## Introduction

Microtubules (MTs) are fundamental to many cellular processes, such as intracellular transport, cell shape and organisation, polarity, and division. A large number of proteins are associated with MT functions, stabilising or destabilising MTs, guiding their growth or linking them to other cell components (*Akhmanova and Steinmetz, 2008*; *Howard and Hyman, 2003*). Plus end tracking proteins (+TIPs) are a very diverse group of proteins that accumulate at the growing (plus) ends of MTs. They form complex interaction networks that are fundamental to the MT dynamics and signalling properties.

Disruption of plus end regulation in diseases like cancer can lead to abnormal cell division and migration, facilitating disease progression (*Aseervatham, 2020*). Drugs targeting MTs are currently widely used in cancer therapies.

End binding protein 1 (EB1) is a key member of +TIPs networks that autonomously tracks the MT plus ends via its N-terminal CH domain. CH binding to MTs affects MT stability (*Zhang et al., 2015*). EB1 dimerises through the C-terminal EB1c region, which increases EB1 affinity to MTs by the cooperative interaction of the two CH domains in the dimer (*Song et al., 2020*). The EB1c region contains conserved EB1 homology (EBH) domain, which regulates +TIPs networks by recruiting a range of other +TIPs, such as cancer-related proteins APC and microtubule-actin cross-linking factor (MACF), through the interaction with the SxIP motifs (*Gouveia and Akhmanova, 2010*; *Honnappa et al., 2009*; *Kumar and Wittmann, 2012*). These small four-residue conserved motifs are usually located in the intrinsically disordered regions of +TIPs (*Buey et al., 2012*; *Jiang and Akhmanova, 2011*).

The crystal structure of the EB1c dimer in complex with MACF SxIP peptide (*Figure 1A*) showed that it forms a parallel leucine zipper coiled-coil dimer followed by a four-helix bundle of the EBH domain. The SxIP motif fits into a deep hydrophobic pocket of the EBH domain between the C-terminal loop and the long helices of the leucine zipper (*Honnappa et al., 2009*). The conserved Ser forms a network of hydrophobic interactions, and Ile and Pro are embedded into a deep hydrophobic pocket; these interactions were proposed to be the main determinants of the ligand binding. The additional contributions of the non-SxIP residues into the binding have been shown through the systematic single-residue substitution analysis that highlighted the importance of the hydrophobic residues that immediately follow SxIP and the positive charges of the subsequent residues (*Buey et al., 2012*). However, it has not been clear to what extent these residues change the overall affinity.

In the solution structure of free EB1c, the region corresponding to the C-terminal region is unstructured and dynamic, with the binding pocket only partially formed (*Figure 1B*; *Almeida et al., 2017*), leading to the conclusion that the binding of the MACF peptide induces the folding of the C-terminus to complete the binding pocket. However, we found that the 4-residue peptide SKIP fragment of MACF has a very low affinity to EBH and induces only small chemical shift changes in the $^1$H,$^{15}$N-HSQC, compared to an 11-residue MACF SxIP-containing peptide, demonstrating that post-SxIP residues have a critical contribution to the complex formation.

Here, we characterise the mechanism and outline the structural features of EBH interactions with the MACF-derived SxIP peptides using biophysical techniques such as NMR and isothermal titration calorimetry (ITC). We show that the post-SxIP residues induce the folding of the EBH C-terminus, thus making a large additional contribution to the peptide binding. To evaluate this contribution, we propose a two-step 'dock-and-lock' model for the peptide interaction with EBH, where the SxIP motif initially docks into the partly formed binding pocket, which subsequently induces the C-terminus folding. Using systematic modifications of EBH and the peptide, we evaluate the thermodynamic and kinetic parameters of the binding. We dissect the contributions of the different peptide regions into the binding and design a peptide with nanomolar affinity to EBH. Our model can be used for the prediction of the EBH affinity to different natural ligands and as a rationale for the design of small molecules to inhibit EB1 interactions that have a recognised contribution to human neoplastic diseases such as cancer (*Abiatari et al., 2009*; *Liu et al., 2009*).

## Results

### SxIP gives specificity in binding to EB1, but flanking residues strengthen the interaction

For the analysis, we selected an EB1 fragment that was previously used by Honnappa et al. to solve crystal structure of the EBH complex with MACF peptide (*Honnappa et al., 2009*). This fragment has a deletion of eight C-terminal residues (D257-G260) that are dynamic and not involved in the ligand binding, as the authors showed by the NMR analysis. For clarity, we use the EBH abbreviation for the 8-residue truncated fragment (residues D191-G260) and EBH-ΔC for the fragment (D191-G252) with an additional 8-residue C-terminal truncation. While the EBH domain was initially associated with a smaller region N223-G252, the structure of the EBH/MACF complex (*Honnappa et al., 2009*) showed a continuous long coiled-coil integrated with the EBH 4-helix bundle and partial folding of the C-terminal region (*Figure 1A*). Thus, structurally, EB1 is better described as having a single, extended

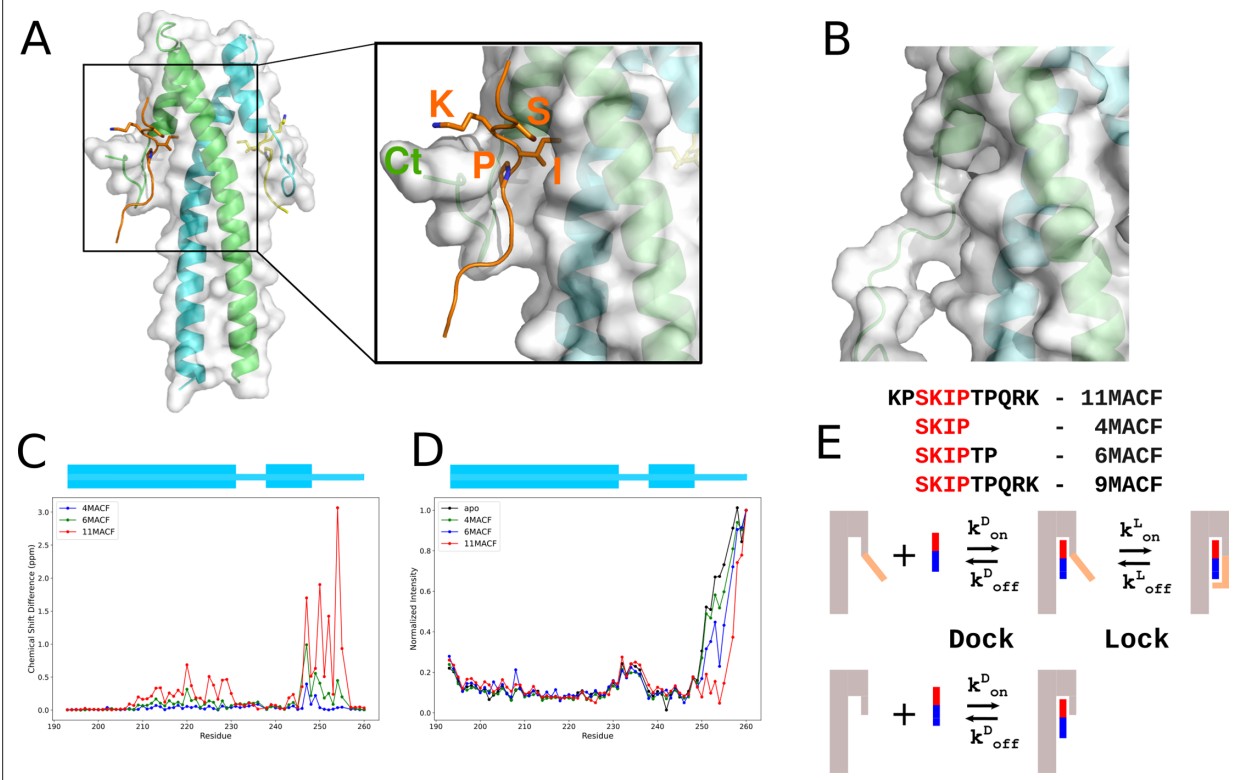

**Figure 1.** Residues that follow the SxIP motif enhance binding by engaging with the C-terminus of the EB1 EBH domain. (**A**) Crystal structure of the EBH domain in the complex with microtubule-actin cross-linking factor (MACF) peptide (PDB ID 3GJO). EB1 is shown as a cartoon with the subunits of the dimer coloured in green and cyan, and a semi-transparent surface. The MACF peptide is coloured orange, with the SxIP motif chains-chains shown as sticks. The zoomed region highlights the binding pocket of EBH formed by the surface of the coiled-coil and the folded C-terminus. (**B**) Zoom on the partly formed SxIP binding pocket in the structure EBH domain free in solution (PDB ID 3EVI) in the same orientation as (**A**). The C-terminal region is unfolded. (**C**) EBH (50 µM) chemical shift changes in the $^1$H,$^{15}$N-HSQC spectra induced by 4MACF (blue, 100-fold excess), 6MACF (green, 100-fold excess), and 11MACF (red, 4-fold excess) peptides. (**D**) Relative intensities of cross-peaks (normalised to the intensity of the C-terminal residue G260) in the $^1$H,$^{15}$N-HSQC spectra of the EBH (50 µM) free in solution (black) and in the presence of 4MACF (blue, 100-fold excess), 6MACF (green, 100-fold excess), and 11MACF (red, 4-fold excess) peptides. (**E**) 'Dock-and-lock' binding model that explicitly considers the role of EBH C-terminus in the interaction with the SxIP peptide. Initially, the binding pocket is partially formed and only contains SxIP-recognition region. Following the initial binding ('dock'), the post-SxIP region of the peptide induces the folding of the of the C-terminus and formation of the full binding pocket ('lock'). The deletion of the EBH C-terminus (orange) removes the 'Lock' stage of the binding, thus reducing the affinity of the interaction. The peptide is shown as a coloured bar, with red corresponding to the SxIP and blue to the post-SxIP regions.

The online version of this article includes the following source data and figure supplement(s) for figure 1:

**Source data 1.** Values of the chemical shift changes in the $^1$H,$^{15}$N-HSQC spectra induced by microtubule-actin cross-linking factor (MACF) peptides for *Figure 1C*.

**Source data 2.** Values of the relative intensities of cross-peaks (normalised to the intensity of the C-terminal residue G260) in the $^1$H,$^{15}$N-HSQC spectra of the EBH complexes with microtubule-actin cross-linking factor (MACF) peptides for *Figure 1D*.

**Figure supplement 1.** Post-SxIP residues increase the binding affinity.

**Figure supplement 2.** Mapping of the largest chemical shift changes induced by the peptide binding on the surface of EB1 EBH domain (**A–C**) and EBH EBH-ΔC (**D–F**) in the complex with the microtubule-actin cross-linking factor (MACF) peptide (green, shown in a stick representation).

**Figure supplement 3.** Deletion of the EBH C-terminus reduces the binding affinity.

EBH domain (residues D257-G260) that contains binding site for SxIP ligands at the interface of the 4-helical and coiled-coil parts of the structure.

Our previous analysis suggested that the folding of the C-terminus induced by the SxIP peptide is required for high binding affinity (*Almeida et al., 2017*). To investigate the roles of different peptide regions in the folding of the C-terminus and the connection between folding and affinity, we compared the binding of the 4- and 6-residue fragments, SKIP (4MACF) and SKIPTP (6MACF), with the 11-residue MACF peptide KPSKIPTPQRK (11MACF) that was used to solve the crystal structure

**Table 1.** Binding parameters of the EB1 homology (EBH) domain interactions with peptides.

| | $K_D$ (µM)* | $\Delta G$ (kJ/mol) | $\Delta H$ (kJ/mol) | $-T\Delta S$ (kJ/mol) | N sites | $k_{off}$ (s⁻¹)[†] | $k_{on}$ (µM⁻¹ s⁻¹) |
|---|---|---|---|---|---|---|---|
| EBH/4MACF | (10400±280) | (–11.312±0.066) | | | | | |
| EBH/6MACF | (1730±240) | (–15.76±0.34) | | | | | |
| EBH/9MACF | 34.00±0.56 | –25.49±0.04 | –24.95±0.06 | –0.52±0.1 | 1.00±0.04 | | |
| EBH/11MACF | 3.5±1.0 (4.91±0.09) | –31.20±0.81 | –39.6±4.5 | 8.4±3.6 | 1.13±0.17 | 130.2±2.1 (143.6±6.0) | 37±11 |
| EBH/11MACF-LLL | 0.287±0.088 | –37.50±0.87 | –35.8±5.3 | –4.9±1.4 | 1.01±0.11 | | |
| EBH/11MACF-VLL | 0.081±0.009 (0.32±0.012) | –40.50±0.25 | –40.2±1.2 | –1.300±0.091 | 0.906±0.056 | 15.63±0.12 (57.5±8.7) | 192±21 |
| EBH/11MACF-VLLRK | 0.016±0.003 | –44.60±0.47 | –27.80±0.30 | –16.80±0.32 | 1.18±0.14 | | |
| EBH/11MACF-VLLRK150 mM NaCl | 0.198±0.012 | –38.5±1.5 | –34.4±1.8 | –4.18±0.17 | 1.03±0.05 | | |
| EBH/13CK5P2 | 0.021±0.022 | –46.4±5.0 | –36.4±8.8 | –9.6±5.0 | 0.876±0.093 | | |
| EBH-△C/6MACF | (7070±1880) | (–12.27±0.66) | | | | | |
| EBH-△C/11MACF | 41.5±8.8 (26.6±0.51) | –24.62±1.7 | –29.1±1.8 | 1.11±0.92 | 1.08±0.35 | 1900±54 | 45.8±9.8 |
| EBH-△C/11MACF-VLL | 18.7±3.1 (25.62±0.82) | –27.00±0.43 | –24.9±4.5 | –2.1±4.8 | 0.99±0.18 | 1541±79 | 82±14 |
| EBH-△C/11MACF-VLLRK | 6.29±0.47 | –29.70±0.17 | –24.6±4.6 | –5.1±4.6 | 0.960±0.056 | | |

All experiments were conducted in the buffer containing 50 mM phosphate (pH 6.5), 50 mM NaCl, 0.5 mM TCEP, and 0.02% NaN₃. Salt concentration was increased to 150 mM NaCl for EBH/11MACF-VLLRK (marked in the table) to test the ionic strength dependence of the binding.

*$K_D$ is determined by ITC and NMR (values in brackets). Average over three technical replicates and standard deviation is reported for ITC. The dissociation constants for 4MACF and 6MACF were determined by fitting chemical shift perturbations; average values over the peaks with the largest perturbations and standard deviation are reported. For other peptides, the NMR binding parameters were determined by the lineshape analysis in TITAN software; the results of the global fit over the well-resolved signals with the largest chemical shift perturbation (**Figure 4—figure supplement 3**) and standard deviation from the fit are reported.

[†]$k_{off}$ is determined by the NMR lineshape analysis (standard deviation reported) and CEST (values in brackets with standard deviations from the fit).

of the complex (**Honnappa et al., 2009**). These peptides induced progressively larger changes in the NMR spectra of the ¹⁵N-labelled EBH domain (**Figure 1C**, **Figure 1—figure supplement 1**). For 4MACF and 6MACF, we observed fast exchange between the free and bound states, characterised by a progressive linear change of the chemical shifts in the ¹H,¹⁵N-HSQC spectra of EBH on peptide addition. This corresponds to a weak EBH interaction with the peptides. Fitting the chemical shift changes into a two-state binding model (**Figure 1—figure supplement 1D and E**) estimates the dissociation constant $K_D$ of the binding as ~10 mM and ~2 mM for 4MACF and 6MACF (**Table 1**), respectively, similar to the $K_D$ values we measured previously for the SxIP-like molecules (**Almeida et al., 2017**). For the 11MACF peptide, many signals are in the slow exchange regime, in agreement with a much stronger interaction ($K_D$ 3.5 ± 1.0 µM from ITC, **Table 1** and **Figure 1—figure supplement 1**). To understand the dramatic effect on the interaction of the non-SxIP residues, we investigated the contributions of these residues to the binding further.

The chemical shift changes in the ¹H,¹⁵N-HSQC spectra for the peptides map to increasingly larger regions on the EBH surface (**Figure 1—figure supplement 2**). Notably, the changes for the 4- to 6-residue peptides are located predominantly in the structured part of EBH, with relatively small effect on the unstructured C-terminal region. In contrast, the 11-residue peptide induced the largest chemical shift perturbations in the C-terminal region (**Figure 1A**, **Figure 1—figure supplement 2A**). Additionally, we observed a different effect on the intensities of the ¹H,¹⁵N-HSQC cross-peaks assigned to the C-terminal region of EBH from the addition of the peptides (**Figure 1D**). In the free EBH, the intensity of the signals of the C-terminal region E251-G260 is much higher than the signals of the coiled-coil part of the protein, as expected for the dynamical C-terminus. In the presence of the 4- and 6-residue peptides, the intensities of these signals still remain much higher than the signals corresponding to the coiled-coil part folded part of EBH, while in the presence of the 11-residue peptide, the intensities of many C-terminal signals became similar to those of the folded part, corresponding to partial immobilisation of the C-terminus. The combination of the chemical shift and intensity changes suggests that the folding of the C-terminal regions seen in the crystal structure of the complex is induced not by the SxIP motif, but by the residues that follow it, and that this folding leads to the much higher binding

affinity. While not conserved in the ligand sequence and not defining ligand specificity (*Honnappa et al., 2009*), the post-SxIP residues clearly have a critical role in the interaction.

The intensities of the last four C-terminal residues of EBH (residues D257-G260) remain high even in the complex with the 11MACF peptide, and this region shows no chemical shift changes. This demonstrates that the final residues of the EBH remain dynamic in the complex and are not involved in the interaction with the ligand. This aligns with the conclusion of *Honnappa et al., 2009*, that the end of the EB1c C-terminal region (residues P261–268) is not required for the ligand binding and can be removed.

The 11MACF peptide has a 2-residue extension at the N-terminus compared to 4MACF and 6MACF. Deletion of these residues in the 9MACF peptide SKIPTPQRK led to the increase of $K_D$ measured by ITC to 34.00 ± 0.56 µM (*Table 1* and *Figure 1—figure supplement 1F*), showing significant contribution of the pre-SxIP region into the binding. Thus, both SxIP flanking regions are important for the binding affinity despite their low conservation, with a dominating effect from the post-SxIP region.

To define the contribution of the EBH C-terminus, we analysed peptide binding to the EBH-ΔC fragment lacking the C-terminal SxIP binding region. For the signals of the EBH-ΔC coiled-coil region, we observed similar changes on peptide addition as in the EBH spectra; however, the amplitudes of the changes were reduced, corresponding to much weaker interactions (*Figure 1—figure supplement 3A–C*). For the 4MACF peptide, the interactions were too weak to estimate $K_D$, while the affinities to 6MACF and 11MACF are reduced 4- and 10-fold, respectively (*Figure 1—figure supplement 3D and E* and *Table 1*). The ITC thermodynamic parameters show that for 11MACF, the reduction in the affinity is caused by the large decrease of the absolute value of the interaction enthalpy, partly compensated by the reduction of the entropy loss (*Table 1*, *Figure 1—figure supplement 3F*). This agrees with the partial loss of the binding pocket and the lack of the additional ordering of the unstructured C-terminus, as the residues that become immobilised in the complex are missing. Notably, chemical shift perturbations in the coiled-coil part of EBH are less extensive for the 11MACF (*Figure 1—figure supplement 2F*), indicating that in the absence of the fully formed binding pocket, the C-terminal region of the peptide has a reduced contact with the coiled-coil region.

The above observations support a 'dock-and-lock' model for the SxIP binding where the SxIP-motif itself only provides the initial recognition, while the full affinity of the interaction is controlled by the folding of the EBH unstructured C-terminal, leading to the additional interactions with the peptide residues immediately following the SxIP motif (*Figure 1E*). To validate this model, we analysed the solution structure of the EBH/MACF complex and measured the interaction dynamics for a series of peptides.

## Peptide binding changes the structure and dynamics of the EBH C-terminus

Following the 11MACF binding, chemical shifts of the EBH C-terminal signals change dramatically and their intensity becomes comparable to the signals on the structured regions of EBH (*Figure 1C and D*). To relate the changes in the spectra to the structural rearrangement on binding in solution, we solved the NMR structure of EBH following the methodology described in *Almeida et al., 2017*. For the coiled-coil EBH region that is structured in the free form (leucine zipper and the 4-helix bundle), we observed a similar pattern of intra- and inter-molecular contacts in the complex, and similar values of the dihedral angles derived from the $^{13}$C-chemical shifts as for the free EBH. In contrast, a large number of new intra-molecular contacts were detected in the C-terminal regions, as well as the inter-molecular contacts between the 11MACF peptide and EBH. This directly demonstrates that the peptide binding has a limited effect on the overall fold of EBH, and most of the structural changes occur in the C-terminal region.

The solution structure determined from the NMR data shows that EBH in the complex with 11MACF has a coiled-coil structure consisting of a leucine zipper and a four helical bundle in the upper region that is practically identical to the structure of the free form (*Figure 2A*). The C-terminus folds over the peptide (*Figure 2B*), forming a loop similar to the loop observed in the crystal form (*Honnappa et al., 2009*). Only a small number of contacts were detected for the N-terminal KPSK region of the peptide, including Ser5477, that shows a single distance contact to Phe218. This suggests a limited role of this region in the interaction with EBH. Interestingly, an extensive H-bond network involving Ser5477 has been identified in the crystal structure (*Honnappa et al., 2009*). The lack of NOE contacts

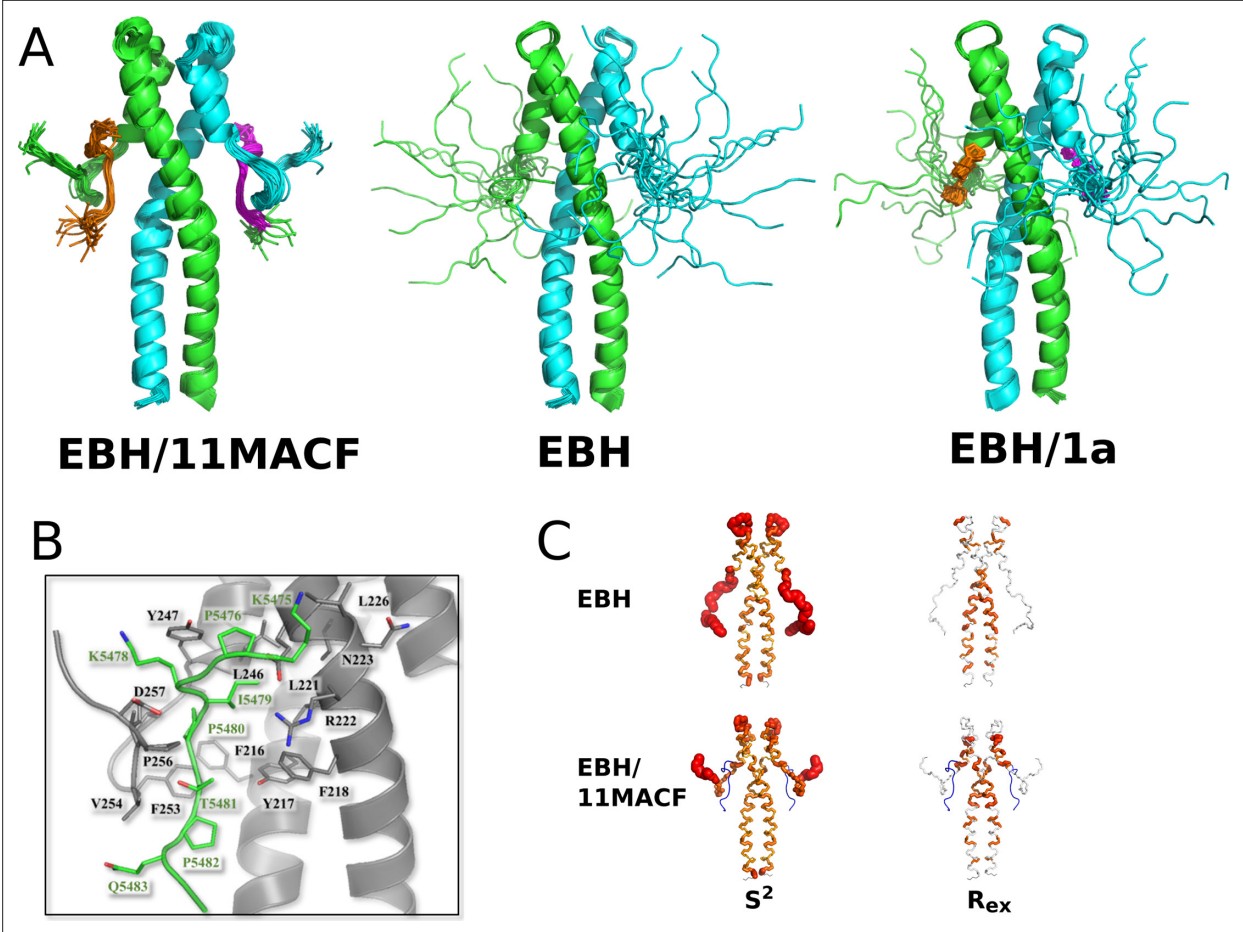

**Figure 2.** Structure and dynamics of the EB1 EB1 homology (EBH) domain. (**A**) Superposition of 20 NMR structures calculated for EBH in the complex with 11MACF peptide (left). Previously reported NMR structures of the free EBH (PDB ID 6EVI, middle) and the EBH complex with a small SxIP-like molecule 1a (PDB ID 6EVI, right) are shown for comparison. The EBH subunits are coloured in green and cyan, the peptide and the small molecule in orange and magenta. (**B**) Representation of the residues forming the contact interface between the EBH and 11MACF peptide, both shown as cartoon, with side chain displayed for all the residues involved in the contacts between the two molecules. EB1 is coloured in grey and 11MACF in green, with oxygen shown in red and nitrogen shown in blue. (**C**) Order parameters $S^2$ (left) and exchange contributions into the relaxation rate $R_{ex}$ (right) calculated from the relaxation parameters for the free EBH (top) and EBH in complex with 11MACF peptide (bottom) mapped on the EBH solution structure. The thickness of the tube is proportional to 1 $S^2$ (left) or $R_{ex}$ (right).

The online version of this article includes the following source data and figure supplement(s) for figure 2:

**Figure supplement 1.** Relaxation parameters measure at 600 (blue) and 800 (orange) MHz for free EB1 homology (EBH) (left) and EBH in the complex with 11MACF (right).

**Figure supplement 1—source data 1.** Values of relaxation parameters for the free EB1 homology (EBH) and EBH complex with 11MACF peptide.

**Figure supplement 2.** Order parameters $S^2$ and exchange rates $R_{ex}$ calculated from the relaxation data for free EB1 homology (EBH) (left) and EBH in complex with the 11MACF peptide (right).

**Figure supplement 2—source data 1.** Calculated $S^2$ and $R_{ex}$ parameters for the free EB1 homology (EBH) and EBH complex with 11MACF peptide.

of Ser5477 suggests that H-bonds in solution are transient and play less significant structural role than has been expected from the crystal structure. The following residue, Lys5478, is solvent exposed and has no NOE contacts with the helical part of EBH. In contrast, the side-chain Ile5479 shows the largest number of NOE distance contacts to the residues of coiled-coil part of EBH, including Tyr217, Phe218, Leu221, Arg222, Ile224, Glu225, Leu246, and Tyr247. The side chains of these residues form a deep hydrophobic pocket that matches the shape of Ile. Similarly, Pro5480 shows hydrophobic contacts to Tyr217, Phe218, and further down with Thr249. Both Pro5480 and Thr5481 engage with the lower region of the 4-helix bundle, just below the SxIP binding site, with NOEs to the aromatic patch [216]FYF[218].

Additionally, the $^{5480}$PT$^{5481}$ part of the peptide makes numerous contacts with the folded hydrophobic region $^{253}$FVIP$^{256}$ of the C-terminus (*Figure 2B*). These interactions position the C-terminus in the linear conformation on top of the peptide, protecting it from the solvent. The aromatic ring of Phe253 in the EBH C-terminus slots between the two Pro residues of the peptide and stacks against the ring of Phe261 of the coiled-coil region, anchoring the folded loop to the EBH core. This is the only direct interaction between the C-terminus and the EBH core; in the absence of the peptide, it would be insufficient to stabilise the structure of the C-terminus. This agrees with the lack of EBH C-terminal folding in the complexes with the shorter MACF peptides (see above).

To assess the dynamic properties of the loop in the free EBH and the peptide complex, we measured $^{15}$N relaxation at 600 and 800 MHz NMR fields, which gives information on both slow and fast exchange processes. At both magnetic fields, the relaxation parameters showed matched sequence distribution profiles (*Figure 2—figure supplement 1*). The elevated R2/R1 ratios and NOE values, corresponding to the low internal mobility, are located in the helical regions of EBH structure, while the reduced values, corresponding to the high dynamics, are in the loop between the helices and the EBH C-terminus (*Figure 2—figure supplement 1*). The order parameter S$^2$ that quantitatively characterises the fast motion and the exchange contribution into the relaxation R$_{ex}$ that characterises slow structural rearrangements (*Figure 2—figure supplement 2*) has been derived by the model-free analysis in the Relax software (*d'Auvergne and Gooley, 2008*). Mapping of these values on the EBH structure (*Figure 2C*) shows that in the free EBH, the whole C-terminus is highly dynamic, in agreement with the variations in the NMR structures and high intensities of the $^1$H,$^{15}$N-HSQC cross-peaks. Surprisingly, high dynamics is also observed in the loop between the helices that is well determined in the NMR structures. This discrepancy is likely caused by the multiple conformations of the loop leading to partially contradicting NMR restraints. In such cases, the NMR structures represent an averaged conformation. In the EBH/11MACF complex, only the last four residues are dynamic; the rest of the C-terminus is fully immobilised. Notably, the S$^2$ values of the C-terminal loop in EBH (*Figure 2—figure supplement 2*) are very close to the values in the helical parts, demonstrating that the structure of the C-terminus in the complex is rigid despite the high initial dynamics of the C-terminus of free EBH and the free peptide in solution. Surprisingly, the low values of R$_{ex}$ show the lack of slow conformational changes in the immobilised C-terminus that are often observed in similar systems. Once formed, the complex is very rigid, suggesting well-matching structures of the peptide and the EBH C-terminus.

The structural and dynamics analysis supports a two-step model of the SxIP peptide binding (*Figure 1E*). In the free form, only the SxIP-recognition part of the binding pocket is formed that allows the initial docking of the peptide predominantly through the hydrophobic interaction of Ile5479. These interactions are insufficient for strong binding because they involve only a small number of the peptide residues. The initial docking induces the folding of the EBH C-terminus through the complementarity between the hydrophobic region $^{253}$FVIP$^{256}$ of the C-terminus and the peptide residues that immediately follow the SxIP motif. The folded C-terminus is fixed to the coiled-coil region through the aromatic stacking of Phe253 and Phe261. The additional interactions induced by the folding of the C-terminus extend the binding pocket and increase the affinity ~20-fold, based on the EBH C-terminal deletion experiments. Next, we used the binding model to design peptides with enhanced binding affinity and to dissect contributions from different peptide regions to the binding.

## Peptide optimisation in the post-SxIP region

The folding of the EBH C-terminus after the peptide binding brings the highly hydrophobic region $^{253}$FVIP$^{256}$ of EBH into contact with a relatively hydrophilic $^{5481}$TPQ$^{5483}$ region of the peptide (*Figure 3A*). We reasoned that the affinity of the interaction can be enhanced by making this area of the peptide more hydrophobic. To assess a suitable substitution, we docked three versions of the modified peptides into the EBH structure of the complex with MACF: KPSKIPLLLRK (11MACF-LLL), KPSKIPVLLRK (11MACF-VLL), and KPSKPILLRK (11MACF-ILL) (*Figure 3—figure supplement 1*). All variants showed a high complementarity to the binding site. The highest fitness score was obtained for 11MACF-LLL, followed by 11MACF-VLL. The score of 11MACF-ILL was similar to that of the wild-type. Based on the docking, KPSKIPLLLRK and KPSKIPVLLRK were selected for the experimental analysis.

The ITC titration curves (*Figure 3C*, *Table 1*) of the WT and the mutants had a sigmoid curve with a single inflection point and fitted to a single site model without any systematic deviation between

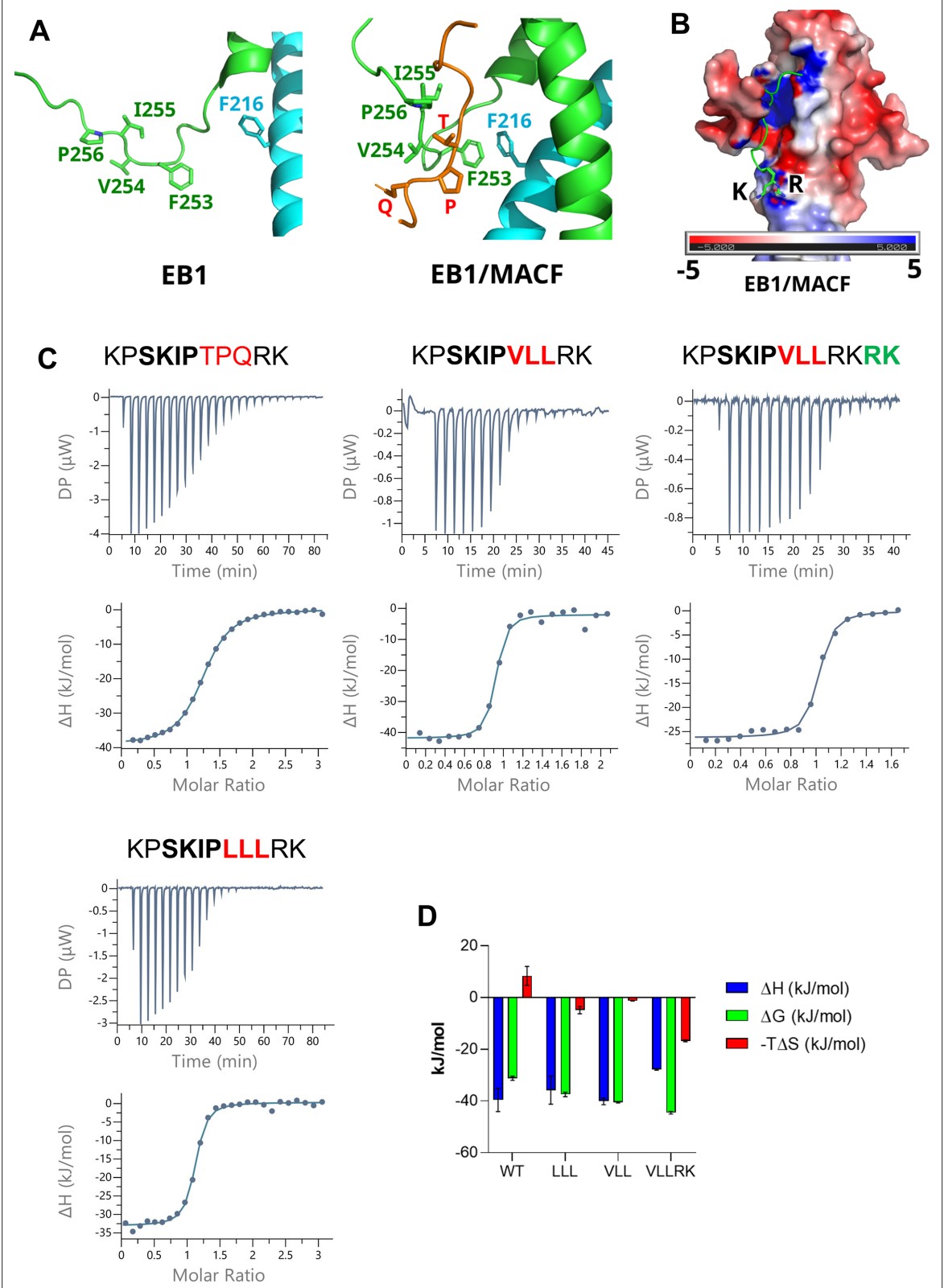

**Figure 3.** Enhancement of the peptide binding through the substitution of the post-SxIP residues. (**A**) Folding of the C-terminus brings the hydrophobic EB1 homology (EBH) residues of that region into contact with the TPQ region of the peptide. Structure of the free EBH (left) and EBH in the complex with 11MACF (right). (**B**) Proximity of the positively charged RK residues of the peptide to the negatively charged patch on the EBH surface. The peptide is shown as a cartoon, RK side chains are shown in a stick representation. Positive and negative electrostatic surface potential is represented by

*Figure 3 continued on next page*

*Figure 3 continued*

blue and red colour, respectively. (**C**) The isothermal titration calorimetry (ITC) titration (top) and the binding isotherm fitted into a single-site binding model (bottom) of the ITC binding experiments for the 11MACF peptide and the mutants 11MACF-LLL, 11MACF-VLL, and 11MACF-VLLRK. The peptide sequences are shown above the graphs, with the changed regions highlighted by the red (mutation) and green (insertion) colours. (**D**) The thermodynamic parameters ΔG (green), ΔH (blue), and −TΔS (red) calculated from the isothermal titration calorimetry (ITC) data.

The online version of this article includes the following source data and figure supplement(s) for figure 3:

**Source data 1.** Values of the thermodynamic parameters for *Figure 3D*.

**Figure supplement 1.** Best scored docking poses obtained for (**A**) 11MACF-WT, (**B**) 11MACF-LLL, (**C**) 11MACF-VLL, and (**D**) 11MACF-ILL.

the fitted curve and the experimental data. The stoichiometric molar ratio was close to one peptide molecule to one EB1c monomer, in agreement with two peptide binding sites in the EB1c dimer and no cooperativity between them. We observed a 10-fold increase in binding affinity for 11MACF-LLL and a further 3-fold increase for 11MACF-VLL, when compared with the wild-type 11MACF. Notably, the enthalpy of the interaction of the 11MACF-LLL variant is lower than for the WT and 11MACF-VLL (*Figure 3D*), suggesting a less optimal match to the binding pocket. This is compensated by a favourable change in the entropy, implying an increased hydrophobic effect, as predicted by the modelling. The binding enthalpy for 11MACF-VLL was similar to the WT, indicating a good fit to the binding pocket. The change in the entropy is less favourable than for 11MACF-LLL, agreeing with the reduced hydrophobicity. However, the net change in the negative free energy is larger for 11MACF-VLL, explaining the affinity increase.

The positively charged RK region at the C-terminus of the peptide does not make any stable contacts with EBH. However, this part of the peptide is close to the negatively charged patch on the EBH surface (*Figure 3B*). We speculated that this proximity creates a favourable electrostatic interaction, enhancing the affinity of the peptide. To test this prediction, we duplicated the RK sequence at the C-terminus in the extended peptide KPSKIP<u>VLL</u>RK<u>RK</u>(11MACF-VLLRK). In agreement with the prediction, ITC showed a further 5-fold increase in affinity, leading to $K_D$ of 16 ± 3 nM (*Table 1*). Overall, through the rational sequence modification, we enhance the affinity by nearly two orders of magnitude, changing the binding from the micromolar to nanomolar range.

Comparison of the $^1$H,$^{15}$N-HSQC spectra of EBH in the free form and different EBH complexes reflects the structural changes associated with the changes in affinity (*Figure 4A*, *Figure 4—figure supplement 1*). The binding of the WT 11MACF leads to the large changes in the positions of the signals of the residues in the binding site through the direct contact. Additionally, the peptide induces the folding of the C-terminus, introducing even larger changes to the corresponding signals. Because of the extensive interaction between EBH and the peptide, nearly all $^1$H,$^{15}$N-HSQC signals change position in the complex. Modification from TPQ to LLL results in further large chemical shift changes, primarily in the signals of the folded hydrophobic part of the C-terminus ($^{253}$FVIP$^{256}$) that interacts with the modified peptide region, and of Phe218 that forms an aromatic stacking contact with Phe253. No significant changes are observed in the signals of the residues forming the SxIP binding site. This suggests structural rearrangement of the C-terminus to match the modified, more hydrophobic peptide sequence. On the LLL to VLL change in 11MACF-VLL, the spectral changes are much smaller and in the same C-terminal EBH region, demonstrating further, small structural rearrangements. The addition of the positively charged RK residues in 11MACF-VLLRK has a minimal effect on the spectra, indicating that the contacts of the charged residues are transient.

Taking ITC and NMR data together, the described peptide modifications optimise the property of the post-SxIP region for high affinity. The residues immediately following SxIP are engaged in the hydrophobic interactions and have to match the complementary FVIP region of EBH. This region is initially unstructured, allowing it to fit different peptide sequences. The closer fit would increase the affinity; however, the specificity of these interactions is limited because the flexible C-terminus can adapt to different sequence. This is reflected in the low sequence conservation beyond the SxIP motif (*Figure 4—figure supplement 2*). Additional, smaller contributions to the affinity come from the non-specific charge interactions between the positive C-terminus and the negative surface of EBH. The separation of the specificity and affinity of the peptide binding shows that different sets of interaction control each stage of the dock-and-lock model (*Figure 1E*). Additionally, the transition to the full complex (lock) depends on the kinetics of C-terminus folding induced by the post-SxIP region of

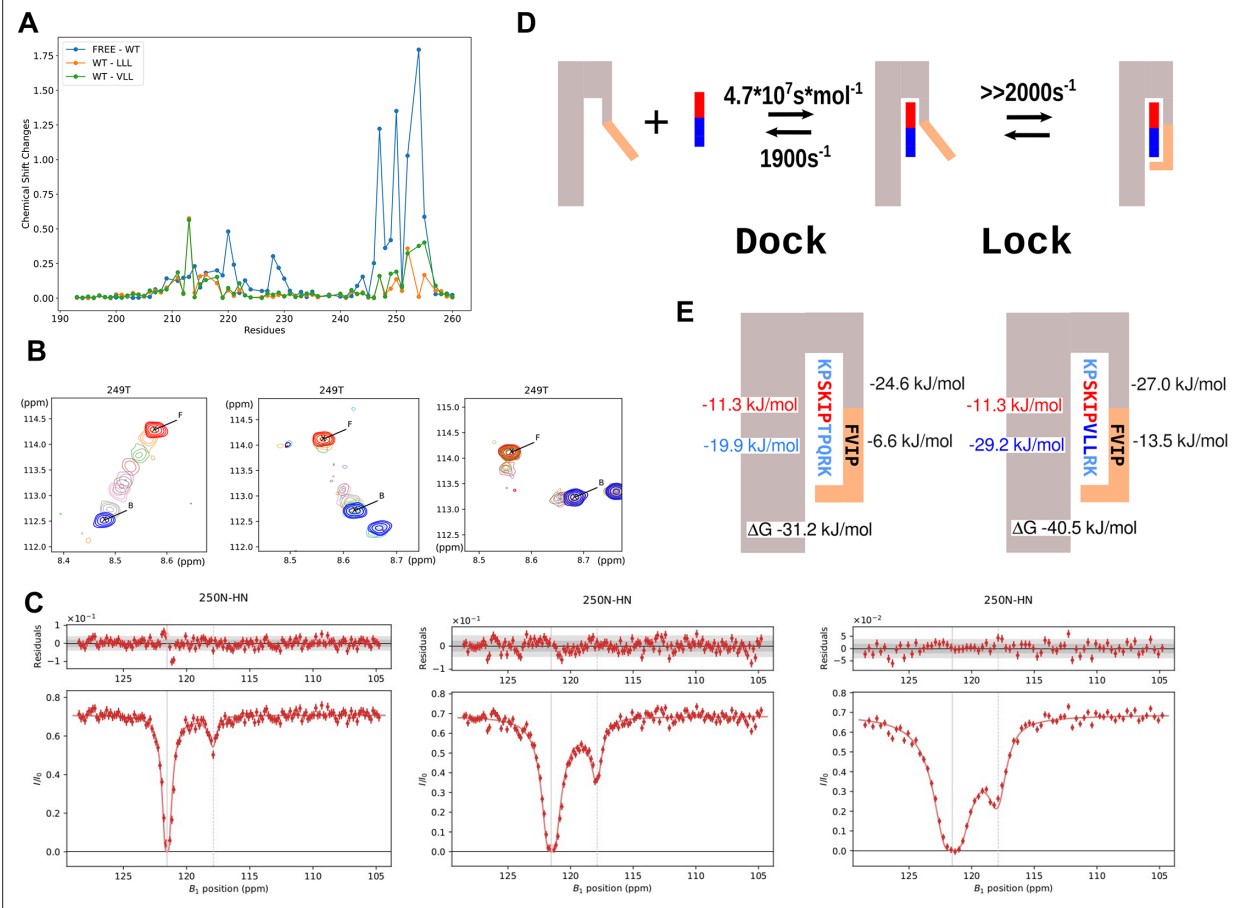

**Figure 4.** Exchange rates and binding energies for the interaction of EB1 homology (EBH) domain with the SxIP ligands. (**A**) 11MACF-LLL and 11MACF-VLL peptides induce increased chemical shift changes compared to the 11MACF. Chemical shift differences in the $^1$H,$^{15}$N-HSQC spectra between the free EBH and the EBH/11MACF complex (blue), EBH/11MACF and EBH/11MACF-LLL (orange), and EBH/11MACF and EBH/11MACF-VLL (green). (**B**) Chemical shift changes in the $^1$H,$^{15}$N-HSQC spectra on peptide addition observed for the EBH interactions with different peptides illustrated for the Thr$^{249}$ signal. Superposition of the spectra for the titration of the EBH-ΔC with 11MACF (left), EBH with 11MACF (middle), and EBH with 11MACF-VLL (right). Signals of the free EBH are shown in red, fully bound EBH in blue, and the intermediate titration points are shown in the pale colours. Notice two additional signals that are observed for the EBH/11MACF-VLL titration at the intermediate concentrations corresponding to the non-symmetrical form where EBH dimer binds a single peptide. These signals can only be observed when the exchange between the different forms is very slow. (**C**) Example of the chemical exchange saturation transfer (CEST) profiles measured for the EBH/11MACF interaction at the irradiation field strength of 12.5, 25, and 50 Hz (left to right) for Asp$^{250}$. Solid curve represents the fitting of the data into the global exchange model with the dissociation rate 143.6 s$^{-1}$ calculated with Chemix software. (**D**) Exchange rates calculated for the EBH interaction with 11MACF peptide from the combination of the NMR data using the two-stage interaction model, where the folding of the EBH C-terminus follows the peptide binding. (**E**) Free energy contributions into the EBH interaction with the 11MACF peptide (left) and the mutated 11MACF-VLL peptide (right). The SxIP motif itself contributes approximately half of the binding energy (–19.9 kJ/mol), with the second half created by the interaction of the KP and TPQRK regions. The C-terminal EBH region that folds on the peptide contributes –6.6 kJ/mol into the binding. The VLL mutation increases the overall contribution of non-SxIP residues into the binding energy to –28.4 kJ/mol, and the binding energy of the EBH C-terminus to –13.5 kJ/mol.

The online version of this article includes the following source data and figure supplement(s) for figure 4:

**Source data 1.** Values of the chemical shift changes in the $^1$H,$^{15}$N-HSQC spectra induced by microtubule-actin cross-linking factor (MACF) peptides for *Figure 4A*.

**Figure supplement 1.** Progressive changes in the HSQC spectra in the complexes with the mutant peptides.

**Figure supplement 2.** Sequence alignment and interactions of SxIP motifs.

**Figure supplement 3.** Progressive changes in the $^1$H,$^{15}$N-HSQC spectra on peptide addition observed for different EB1 interactions that were used to evaluate the exchange rates with TITAN software for the corresponding EB1 homology (EBH) complexes.

**Figure supplement 3—source data 1.** Values of the chemical exchange saturation transfer (CEST) measurements for EBH/11MACF complex for *Figure 4—figure supplement 3A*.

**Figure supplement 4.** Chemical exchange saturation transfer (CEST) profiles for the residues with the $^{15}$N chemical shift differences between the signals of the free and the bound states larger than 1 ppm.

the peptide. To evaluate the kinetic parameters and to develop the binding model further, we investigated the kinetics of the EBH interaction with WT and modified MACF peptides.

## Kinetics of the target peptide recognition by EBH

NMR spectroscopy provides a diversity of methods to measure exchange processes at a wide range of timescales (*Furukawa et al., 2016*; *Camacho-Zarco et al., 2022*). The $^{15}$N relaxation analysis (see above) demonstrates the lack of fast and slow conformational changes in the folded EBH regions both in the free form and in the complex with 11MACF (*Figure 2C*). This suggests a low population of the intermediate 'dock' state and/or fast transition to the final 'lock' state of the binding model (*Figure 1E*). This conclusion is further supported by the lack of the exchange contribution in the relaxation dispersion experiments even for the residues with the largest chemical shift changes of ~2000 Hz (data not shown). This prevents us from the direct evaluation of the kinetic parameters for the exchange between the 'lock' and the 'dock' stages. However, the parameters for the initial peptide docking can be measured with the truncated EBH-ΔC, where the C-terminal hydrophobic residues that contact post-SxIP region of the peptide are removed. Then, the kinetic parameters of the second 'lock' stage can be estimated from the overall exchange rate in the presence of the C-terminus.

The effect of the chemical exchange is clearly manifested in the exchange broadening of the EBH signals in the $^{1}$H,$^{15}$N-HSQC spectra in the course of the titrations (*Figure 1—figure supplement 1*, *Figure 4—figure supplement 3*). To evaluate the exchange rates, we fitted the spectral changes into the exchange model using TITAN software (*Waudby et al., 2016*). Large changes in the EBH chemical shifts that are observed in many signals on binding allow probing a wide range of the exchange rates for different regions of EBH (see Appendix 1 for more information on the TITAN analysis).

For the quantitative binding analysis in the TITAN software, we selected non-overlapped signals with clear changes in the peak amplitude and position in the course of the titration (*Figure 4B*, *Figure 4—figure supplement 3*). When spectral changes were fitted independently for each residue, the $K_D$ values and exchange rates were similar, indicating that binding kinetics is similar for different regions of the binding site. We therefore used a global fitting of all signals to evaluate the overall kinetic parameters of the binding. The $K_D$ values obtained independently from the TITAN lineshape analysis were similar to the value measured by ITC (*Table 1* and Appendix 1 information), validating the NMR analysis. The fitting of the EBH/11MACF and EBH-ΔC/11MACF-VLL data required the use of the dimer binding model. For the WT peptide, this model showed no cooperativity between the two binding sites in the dimer, while a weak negative cooperativity has been detected for the 11MACF-VLL peptide binding. This conclusion was not supported by the ITC data that showed no bi-phase features in the titration profile and consistent fit to the single site model. This demonstrates that the effect of the cooperativity on the binding constant is too small to be detected by ITC and unlikely to have a functional consequence.

The dissociation rate of 130.2 ± 2.1 s$^{-1}$ evaluated for the EBH/11MACF interaction with TITAN is optimal for the chemical exchange saturation transfer (CEST) NMR experiments, where saturation is transferred from the high-population free state to the low-population complex (*Vallurupalli et al., 2012*). These experiments measure the exchange rate directly and are conducted at the low ligand concentrations, where the population of the fully bound EBH dimer with two peptides is much lower than that of a single peptide complex. This makes the indirect allosteric effect on the binding negligible and simplifies the binding model, thus providing direct information on the peptide interaction with the free protein. The large change of the $^{15}$N chemical shifts for the residues in both the preformed part of the binding pocket (residues 220, 228, 247, 248, and 249) and the dynamic C-terminus that folds on the peptide binding (residues 250, 252, 255, and 255) allows independent measurement of the kinetic parameters for these two regions. As expected, we detected a strong CEST effect for all the residues with sufficiently large changes in the $^{15}$N chemical shifts (*Figure 4C*, *Figure 4—figure supplement 4A*). Independent fitting of the CEST profiles for the individual residues (*Figure 4—figure supplement 4B*) gave very similar dissociation rates for all the residues. We therefore obtained the overall exchange rates by simultaneous global fitting of all CEST profiles (*Figure 4C*, *Figure 4—figure supplement 4B*). The calculated dissociation rate of 143.6 ± 6.0 s$^{-1}$ was in excellent agreement with the lineshape TITAN analysis (*Table 1*). For the weaker EBH-ΔC/11MACF interaction, we did not detect any CEST effect, in agreement with the much faster dissociation rate of 1900 ± 54 s$^{-1}$ evaluated

by the lineshape analysis (*Table 1*). For the stronger EBH/11MACF-VLL interaction, the intensity of the CEST effect was significantly lower than for EBH/11MACF at the same protein:peptide ratio, the exchange rates from the CEST were significantly higher than from the lineshape analysis, and protein:peptide ratio estimated by CEST was several times lower than used in the experiment. This indicates that the exchange rates are too slow for the accurate CEST analysis, in agreement with the low rate of 15.63 ± 0.12 s⁻¹ obtained by the lineshape fitting.

## Model of the target recognition by EBH

Based on the NMR structural and binding analysis, we proposed a two-step 'dock-and-lock' binding model (see above). We can now evaluate the binding parameters for each stage using the exchange parameters derived from the lineshape changes and CEST experiments. The C-terminal deletion of the EBH-ΔC fragment eliminates the 'lock' step and can be used to calculate the parameters for the docking. The fitting of the NMR lineshape changes to a two-state model gives $K_D$ of 26.6 ± 0.51 µM, close to the value of 41.5 ± 8.8 µM from the ITC measurements, and the dissociation rate of 1900 ± 54 s⁻¹ (*Table 1*). Since ITC is generally a more accurate method for the binding constant evaluation, we use the ITC value as the dissociation constant for the dock stage. The on-rate for the docking can then be evaluated as 45.8 ± 9.8 µM⁻¹ s⁻¹, which is similar to the rates reported for diffusion-controlled binding (summarised in *Shammas et al., 2016*). This agrees with the easy access to the open, partly formed binding pocket of EBH.

When analysing the lineshape changes for the full EB1c domain interactions with 11MACF and the mutated peptides, we did not detect separate signals from the intermediate 'dock' stage despite the large chemical shift difference between the free and the bound states and slow-exchange regime. In addition, we did not observe any exchange broadening for the EBH complex formed at the high access of the peptide, where the population of the free state is negligibly low. (Under these conditions, the complex is in the equilibrium between the 'dock' state where C-terminal region unfolded with the chemical shifts similar to the free state and the locked complex with the fully folded C-terminus.) The lack of signals from the 'dock' state and the absence of the exchange contribution from the exchange between the 'dock' and the 'lock' states suggests that the exchange between these states is very fast compared to the chemical shift differences, or the population of the intermediate 'dock' state is very low.

Under the conditions of the fast exchange between the two bound states ('dock' and 'lock'), they can be represented by an average state, and the binding approximated by a two-state model (*Březina et al., 2022*) with the overall dissociation constant $K'_D$ and rate $k'_{off}$ calculated as:

$$K'_D = K^D_D \frac{K^L_D}{K^L_D + 1} \tag{1}$$

$$k'_{off} = k^D_{off} \frac{K^L_D}{K^L_D + 1} \tag{2}$$

where $K^D_D$ and $k^D_{off}$ are the dissociation constant and rate of the dock step, respectively, and $K^L_D$ is the dissociation constant of the lock step. These equations show that the overall dissociation constant $K'_D$ and rate $k'_{off}$ are scaled down proportionally to the fraction of the protein in the intermediate state relative to all bound population. Effectively, the 'lock' step depletes the population of the protein where ligand can dissociate, thus decreasing the overall dissociation constant and dissociation rate. If $K^L_D \ll 1$, corresponding to the case when most of the bound protein is in the final locked state, the overall constants decrease in proportion to the dissociation constant of the lock state.

The parameters of the 'dock' step can be estimated from the peptide interaction with EBH-ΔC that lacks the C-terminal and cannot transfer to the locked state (*Figure 1E*). The dissociation constant of the 'lock' step can then be calculated from *Equations 1 and 2*. Thus, for 11MACF $K^D_D = 41.5 \pm 8.8$ µM and $k^D_{off} = 1900 \pm 54$ s⁻¹, $K'_D = 3.5 \pm 1.0$ µM and rate $k'_{off} = 130.2 \pm 2.1$ s⁻¹, (*Table 1*), giving $K^L_D = 0.08 - 0.07$. This corresponds to only ~8% of the bound population being in the intermediate dock state.

The low population of the 'dock' state could potentially still be sufficient to cause exchange line-broadening under fully bound conditions (large ligand excess) if the exchange rates for the lock process are sufficiently slow. The absence of the broadening suggests fast exchange on NMR timescale, with $k^L_{ex} = k^L_{on} + k^L_{off} \gg 2000\,s^{-1}$ (from the maximum differences in the chemical shifts between

the free and the bound states). With $K^L_{eq} \ll 1$, the exchange is dominated by the on-rate, giving an estimate for $k^L_{on} \gg 2000\, s^{-1}$.

When the population of the intermediate state is low, its contribution to the NMR signals becomes negligibly small, and the measured lineshape changes and CEST saturation transfer are only associated with the exchange between the free and the fully bound 'lock' forms. Under this condition, the overall on-rate of the two-step binding approximated with the two-site model is generally slower than the on-rate of the first dock step (see Appendix 1 for more information):

$$k'_{on} \leq k^D_{on} \tag{3}$$

with the rates becoming equal when the exchange in the second lock step is fast compared to the initial dock step (dock step is rate-limiting).

Using the parameters EBH-ΔC/11MACF binding as the approximation for the first 'dock' step and the EBH/11MACF binding for the overall parameters of the two-step model (**Table 1**), we estimate $k^D_{on} = 45.8 \pm 9.8\,$ μM$^{-1}$ S$^{-1}$ and $k'_{on} = 37 \pm 11\,$ μM$^{-1}$ S$^{-1}$ (**Table 1**), in agreement with **Equation 3**. The similarity of these ratios supports the fast exchange rate of the 'lock' step. The estimated kinetic parameters in the two-step model of the EBH/11MACF interaction are summarised in **Figure 4D**.

For the EBH-ΔC/11MACF-VLL, we observed a ~2-fold decrease in the dissociation constant, mostly associated with the increase in the on-rate to $82 \pm 14\,$ μM$^{-1}$ S$^{-1}$. This could be explained by the additional hydrophobic interaction of the VLL region of the mutant peptide with the hydrophobic surface of the coiled-coil that is easily accessible and may help in steering the peptide docking. Interestingly, the on-rate of the EBH/11MACF-VLL interaction increases further to $192 \pm 21\,$ μM$^{-1}$ S$^{-1}$ on the addition of the EBH C-terminus, contrary to **Equation 3**. This may reflect a more complex binding process. However, a more likely explanation is the difficulty of estimating accurately very slow rates from the lineshape analysis, as the lineshape becomes less sensitive to the variation in the exchange rate when the exchange is very slow. Assuming the fast exchange of the 'lock' step, we can calculate $K^L_D = 0.0043$ from **Equation 1**.

The decrease in $K^L_D$ for 11MACF-VLL compared to 11MACF corresponds to a decrease in the population of the intermediate state from 0.08 to 0.0043. This is expected to cause a small increase in the difference in the chemical shift values between the free and the bound forms for the signals of the C-terminal residues of EBH. In agreement with this prediction, we observed a general systematic increase of the chemical shift differences for the residues at the C-terminus most affected by the peptide binding for 11MACF, 11MACF-VLL, and 11MACF-VLL (**Figure 4A** and **Figure 4—figure supplement 1**). This change supports the fast exchange of the C-terminus between the free and the bound state in the saturated complex, although non-linear shift changes between the complexes also indicate some conformational difference in the complex.

The dissociation constants measured for the different peptides (**Table 1**) allow us to define contributions from different regions into the free energy of the binding (**Figure 4E**). These contributions can be assigned to the 'dock' and 'lock' stages. The short fragments 4MACF and 6MACF do not cause folding of the EBH C-terminus, leading only to a small decrease in ΔG on the increase of the peptide length from –11.312 ± 0.066 to –15.76 ± 0.34 kJ/mol. A much larger decrease in free energy to –31.20 ± 0.81 kJ/mol is observed for the full 11MACF that induces C-terminal folding. Some of this –19.9 kJ/mol total decrease is associated with the two additional N-terminal residues (~–6 kJ/mol, difference between 9MACF and 11MACF peptides), although the majority of the ΔG decrease (~–14 kJ/mol) is caused by the C-terminal extension. This demonstrates that approximately half of the free energy change on the binding is defined by the recognition of the SxIP motif, and the rest by the much less specific binding of the variable sequence that immediately follows SxIP, with a smaller contribution from the variable N-terminal region. The high contribution from SxIP makes the dock stage very sensitive to single-residue mutations in the recognition part of the peptide.

The folding of the C-terminus reduces ΔG by ~–6.6 kJ/mol, released in the 'lock stage' of the interaction (the difference between EBH/11MACF and EBH-ΔC/11MACF binding energies), bringing the overall $K'_D$ from high- to low-micromolar range. The TPQ sequence of 11MACF is not ideal for the interaction with the hydrophobic EBH C-terminus. Replacing this with a hydrophobic VLL sequence improves the match between the peptide and the EB1 C-terminus and reduces free energy from the interaction between the peptide and EB1 C-terminus by an additional ~–9.3 kJ/mol. Notably, the dependence of the binding constants of interactions of EBH-ΔC on peptide length and composition

shows that a most significant contribution to the binding comes from the interaction between post-SxIP residues and the coiled-coil region of EB1c domain, although the pre-SxIP residues also have a significant effect.

Further reduction of the free energy of ~–4.1 kJ/mol is possible by adding positively charged residues to the peptide C-terminus that interact non-specifically with the negatively charged surface of EBH. Strong contribution from the electrostatic interaction is supported by a large increase in the dissociation constant on the increase of the salt concentration from 50 to 150 mM for the EBH/11MACF-VLLRK interaction from 16 ± 3 nM to 198 ± 12 nM (*Table 1* and *Figure 4—figure supplement 2B*).

The variation of the sequence in the post-SxIP region observed in the EBH ligands (*Figure 4—figure supplement 4C*) is expected to affect strongly the affinity of the SxIP ligands, with more hydrophobic sequences binding stronger. To test this, we measured the affinity of the SxIP peptide from CK5P2 (KKSRLPRILIKRSR) that has a C-terminal sequence that is similar to the 13MACF-VLLRK mutant. As predicted, the 13CK5P2 had $K_D$ = 21 ± 2.2 nM that is very close to that of the mutant.

Overall, the change in the free energy on the complex formation consists of three similar contributions: SxIP binding to the partially formed binding pocket, non-specific interactions between the post-SxIP region and the folded part of EBH and folding-induced interaction between the unstructured EBH C-terminus and post-SxIP region of the peptide. Additional, smaller, contribution comes from the N-terminal residues immediately preceding SxIP.

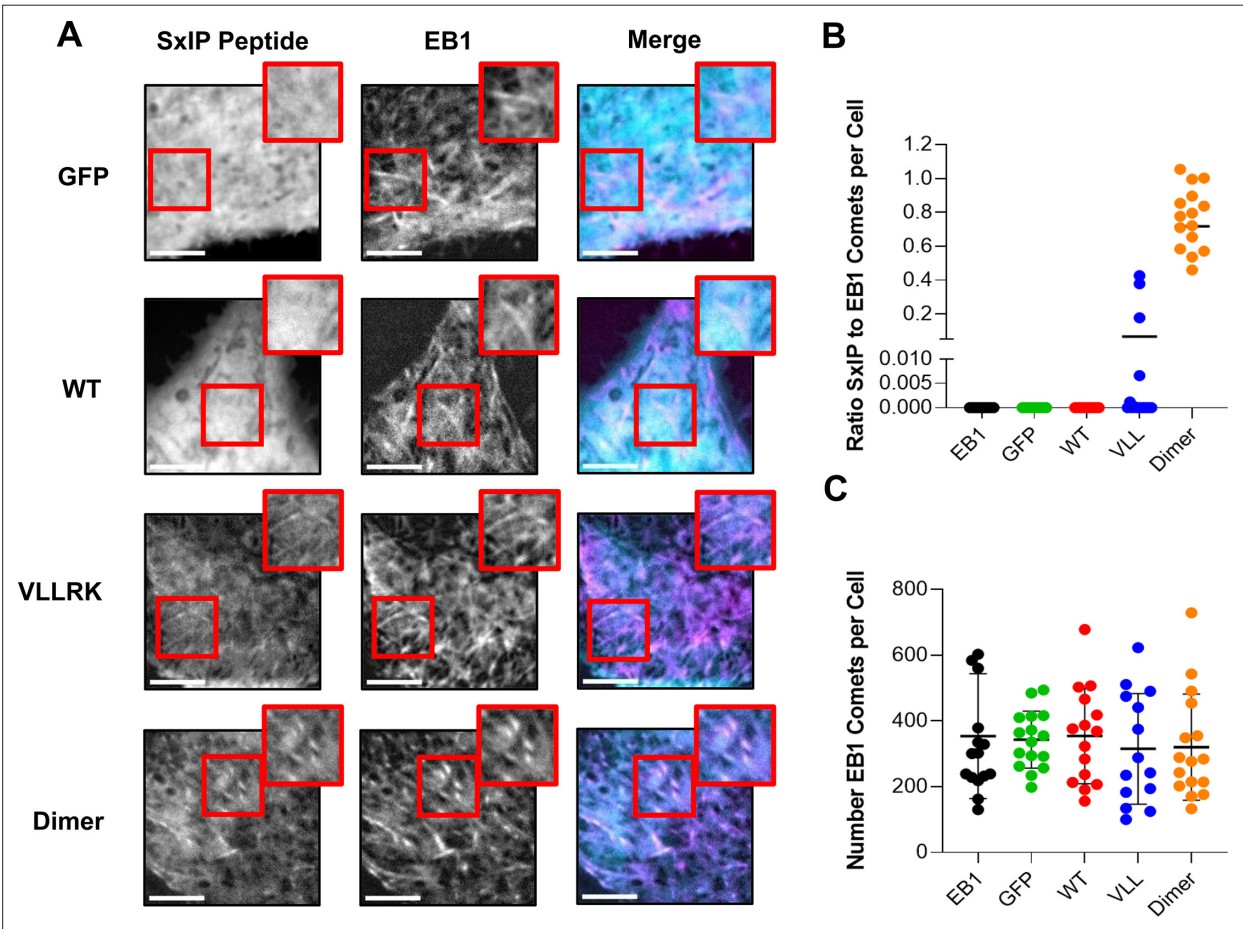

**Figure 5.** High-affinity peptides form comet-like structures that colocalise with EB1 comets. (**A**) Live images of HeLa cells transiently transfected with SxIP peptide constructs and EB1 plasmid. Images represent averaging three frames at 1 s intervals. Insets represent zoomed-in regions of the cell to depict SxIP peptides at EB1 comets. Scale bar: 5 µM. (**B**) Ratio of SxIP comets to EB1 comets present in a cell. (**C**) EB1 comet number per cell. Lines represent the mean with error bars representing the standard deviation. Changes in comet number with each peptide were not significant when t-test was used with the EB1 construct-only control (EB1). All experiments were performed to an N=3.

The online version of this article includes the following source data for figure 5:

**Source data 1.** Values of the ratios of SxIP to EB1 in the comets and comet numbers for *Figure 5B and C*.

## High affinity peptides localise into comet-like structures in cells

Through the rational optimisations of the peptide sequences, we generated SxIP peptides with the enhanced affinity to EB1. To relate their affinity to the EB1-dependent recruitment to the plus ends of MTs, we inserted them into an mTFP containing mammalian vector and followed their ability to localise into mCherry-EB1 comets in HeLa cells. For the WT peptide ($K_D$ = 3.5 ± 1.0 µM), we observed a ubiquitous cytosolic localisation with little to no peptide localised into comets (*Figure 5A*), similar to that of the GFP control. Close inspection of images and movies at high contrast revealed a small number of comets enriched in the peptide that were too faint for the quantitative analysis, comparable, although lower, to the reported previously (*Honnappa et al., 2009*). A measurable localisation to comets could be detected with the MACF-VLLRK peptide ($K_D$ = 16 nM). However, the ratio of peptide to EB1 comets was extremely low at around 0.02 (*Figure 5B*), showing that only a small number of comets contained detectable amounts of the peptide. This ratio dramatically increased to ~0.8 with the dimeric MACF-LZ-VLLRK peptide, showing that only a dimer localised to almost every plus end comet we traced. EB1 comet number remained the same with all peptide transfections (*Figure 5C*). These experiments suggest that the enhanced affinity increases the localisation of ligands to MTs, but even nanomolar affinity is still insufficient for the recruitment of the monomeric ligand.

Notably, the number of comets we detected with our monomeric constructs was significantly smaller than reported by *Honnappa et al., 2009*. Considering a very strong effect of dimerisation on the localisation of the SxIP peptides, this is likely to be due to the ability of EGFP used in their study to form weak dimers that is absent in our mTFP fluorescent tag.

## Discussion

The interaction of EB1 with the SxIP motif-containing proteins is fundamental to the assembly of signalling complexes at the plus ends of MTs. These complexes regulate the assembly and stability of MTs themselves, guide MT growth, and link MTs to other cellular processes, such as cell division or assembly of adhesion complexes (*Akhmanova and Steinmetz, 2008*; *Howard and Hyman, 2003*). In this study, we demonstrated that while the SxIP motif is the only conserved feature of the SxIP-containing proteins, it alone is insufficient for the high-affinity interaction. By comparing the interaction of the EBH domain of EB1 with a range of SxIP-containing peptides, we identified the critical role of the post-SxIP region in the complex formation. These residues induce folding of the EBH unstructured C-terminus following the initial docking via the SxIP motif, which increases affinity by several orders of magnitude. Thus, the overall binding of the SxIP-containing ligands to EBH can be described as a two-step dock-and-lock process, with the initial docking through SxIP motif providing the specificity of the interaction, and the subsequent folding of the EBH C-terminus defining the full binding affinity (*Figure 4D and E*). We evaluated the kinetic parameters of the binding and the contributions from different regions into the binding energy and used the model to increase the affinity of the SxIP peptide ~100-fold through mutations of the post-SxIP region. The mutated peptide showed enhanced dynamic recruitment to the plus ends of MTs in cells, supporting the functional role of the post-SxIP region. We identified previously uncharacterised SxIP-containing EBH ligand CK5P2 (KKSRLPRILIKRSR) with the post-SxIP region that is similar to our high-affinity mutant 13MACF-VLLRK and demonstrated that it has similarly high-affinity $K_D$ of 21 ± 2.2 nM. Overall, our results agree and extend the reported analysis of the substitutions of the individual residues in the SxIP ligands (*Buey et al., 2012*) and provide a dynamic model for the target recognition by EB1.

Our dock-and-lock model is a variant of the induced fit that is a dominant binding mechanism of IDPs (*Sugase et al., 2007*; *Wright and Dyson, 2009*; *Mollica et al., 2016*). We used a separate name for our model to highlight the clear structural definitions of each step of the binding and the lack of the internal motions in the complex. Our previously reported structure (*Almeida et al., 2017*) and the relaxation analysis reported here (*Figure 2C*) demonstrate that in the absence of the ligand, the EBH domain has only partially formed SxIP binding site and a dynamic C-terminus. Because of this, the ligands initially interact with only the folded part of EBH, and the folding is induced after the binding, which mechanistically justifies the two-step model. The combination of the NMR data shows that folding of the C-terminus happens too fast, making its direct detection and evaluation impossible. As a result, the overall binding process can be approximated by a simple two-step binding. However, this model does not fit with the mechanistic description of the binding process from the structural

information and does not provide insights into the binding mechanism and contribution from different regions of EBH and the SxIP ligand. We, therefore, adopted a structurally supported two-step binding model for our analysis and used EBH and ligand truncations to evaluate the kinetic parameters of each step and contributions from different regions into the binding.

The SxIP motif itself has an extremely low affinity to EBH as it fails to induce the formation of the full binding pocket through folding the EBH C-terminus. This folding requires post-SxIP residues that immediately follow the SxIP motif. The folding is driven by the hydrophobic interaction of the FVIP region in the dynamic EBH C-terminus that folds on top of the post-SxIP region and locks it in the position through the stacking of two Phe aromatic rings (*Figure 3A*). These interactions completely immobilise the EBH C-terminus and the peptide ligand, making the completed binding pocket as rigid as the rest of the EBH coiled-coil structure (*Figure 2B*).

This type of binding resembles the interaction of HIV-1 protease with group-specific oncogene (Gag), where the protease has dynamic flaps that fold over the bound substrate and trap it in the catalytic site, thus facilitating the catalysis (*Deshmukh et al., 2017*). Similar loop folding has also been observed in other enzymes, such as some classes of sulfatransferases and lipases, where the closed loops shield the substrates from the bulk solvent and prevent non-productive hydrolysis (*Cook et al., 2013*; *Khan et al., 2017*). However, this is rather unusual in the adaptor proteins similar to EB1 function. In the majority of the reported cases, the folding is limited to the initially unstructured ligand (*Morris et al., 2021*). Part of the ligand often remains dynamic in the complex, forming a set of fuzzy interactions with the folded partner (*Sugase et al., 2007*).

The dock-and-lock mechanism of the EBH binding has a number of functional benefits. The open, partly formed binding pocket is easily accessible, leading to the fast on-rate. The initial, partly formed, binding pocket is very small, primarily fitting the Ile side chain. This makes the pocket highly suitable for the recognition of the short SxIP motif. Despite the small size, the pocket has a very distinct shape, leading to a very specific recognition. However, because of the small size, the binding energy is low, resulting in the high off-rate. This explains the high specificity, but low affinity of the SxIP binding.

The additional residues that follow the SxIP motif contribute in two different ways. First, they enhance the affinity through the interaction with the folded EB1c coiled-coil, with the increase of $K_D$ from 2 mM for 6MACF to 41 µM for the 11MACF interaction with EBH-ΔC (*Table 1*). The fast off-rate of $1900 \pm 54$ s$^{-1}$ determined for the 11MACF/EBH-ΔC interaction from the chemical shift changes in the $^1$H,$^{15}$N-HSQC spectra corresponds to a very high on-rate of $45.8 \pm 9.8$ µM$^{-1}$ s$^{-1}$ calculated from the $K_D$ value. This is similar to the reported diffusion-controlled rates enhanced by the electrostatic interactions (*Sugase et al., 2007*; *Shammas et al., 2013*), which agrees with the open binding pocket and contributions of the C-terminal positive charges of the peptide identified with the mutations and strong dependence of the affinity on the salt concentration (*Buey et al., 2012*). Limited chemical shift perturbations outside of the SxIP pocket for the 11MACF/EBH-ΔC interaction suggest that in the absence of the C-terminus, the interactions of the post-SxIP residues with the folded coiled-coil region are transient and have more effect on the on- than the off-rate.

The second effect of the post-SxIP residues is to induce the folding of the EBH C-terminus, which locks the peptide into the fully folded state. While not available from the direct measurements, the rate of this step can be evaluated from the rates of the dock step (above) and the overall rate for the complex formation using the two-step model. Both chemical shift analysis and CEST measurement give consistent values for the overall off-rate of ~130 s$^{-1}$ for the 11MACF/EBH interaction. This corresponds to ~15-fold decrease from the rate $1900 \pm 54$ s$^{-1}$ of the dock step, which is similar to the differences in the overall $K_D$ values for the 11MACF/EBH-ΔC (only dock step) and 11MACF/EBH (both dock and lock steps). The proportional decrease corresponds to the fast exchange condition for the lock step (*Březina et al., 2022*), with forward rate >>2000 s$^{-1}$, estimated from the chemical shift differences. This rate is much higher than the dissociation rate of the dock step. As a result, the binding is highly productive, with practically all encounter complexes proceeding to the fully bound state.

Overall, the binding mechanism of EBH maximises the on-rate through the open binding pocket optimised for the short SxIP domain recognition, making this motif the only feature required for the target recognition. Insufficient binding affinity of the small SxIP-recognition pocket is strongly enhanced by the folding induced by the post-SxIP region. The main requirement for this region is to support the hydrophobic interactions with the EBH C-terminus. These interactions are much less specific as the dynamic C-terminus can adjust and accommodate the changes. This allows for significant sequence

variation and explains the high conservation of the SxIP motif and lack of the conservation in the post-SxIP region (*Figure 4—figure supplement 2A*). Notably, many IDR ligands have a small number of key hydrophobic residues that are critical for the formation of the encounter complex (*Sugase et al., 2007*). It is not clear yet whether these residues define small recognition motifs for all the IDR ligands, or only some IDR, such as SxIP ligands, have well-defined small recognition motifs.

Based on the understanding of the post-SxIP region contribution to the affinity, we designed an MACF mutant 13MACF-VLLRK with a highly increased affinity from 3.5 ± 1.0 µM to 16 ± 3 nM. In this design, we increased the hydrophobicity of the immediate post-SxIP region that interacts with the EBH C-terminus and introduced additional charges at the peptide C-terminus to enhance the electrostatic steering effect. These substitutions agree with the reported previously systematic single-residue replacement analysis (*Buey et al., 2012*), extending it to the multi-residue substitutions. We found that some of the SxIP ligands contain hydrophobic post-SxIP residues, similar to our high-affinity mutant (*Figure 4—figure supplement 2A*), and validate high-affinity interaction of one of them, CK5P2, with $K_D$ of 21 ± 2.2 nM (*Table 1* and *Figure 4—figure supplement 4C*). This demonstrates the scope of the affinity modulation through the variation in the post-SxIP regions of different EBH ligands that can be identified from the sequence comparison.

Endogenous EB1 forms dynamic condensates via phase separation that track plus ends of MTs in comet-like structures (*Song et al., 2023*). These structures recruit many of the SxIP-containing +TIPs proteins such as MACF and APC (*Kita et al., 2006*; *Honnappa et al., 2009*) and regulate MT polymerisation, catastrophe, and rescue (*Vaughan, 2005*). The SxIP +TIP proteins effectively localise to EB1 comets (*Akhmanova and Steinmetz, 2015*) and condensates (*Song et al., 2023*), and enhance condensate formation to a different degree (*Song et al., 2023*). We found that the comet localisation depends on the SxIP-peptide affinity, but even the mutant peptide with the nanomolar affinity does not localise effectively (*Figure 5A*). Only when the peptide was dimerised through the leucine-zipper motif did the number of the SxIP comets become similar to the number of EB1 comets per cell (*Figure 5B*). This shows that the recruitment of the SxIP ligands requires multi-valent interactions that are characteristic of the condensates (*Boeynaems et al., 2018*; *Zumbro and Alexander-Katz, 2021*; *Mohanty et al., 2022*) and is partly driven by the condensation. Notably, most of the SxIP proteins contain additional interaction sites with other +TIP proteins, and some of them have multiple SxIP motifs (*Kumar and Wittmann, 2012*; *Akhmanova and Steinmetz, 2015*). Therefore, effective targeting of the SxIP and other interactions in the +TIP MT networks may require high-affinity multi-valent inhibitors, which should be considered in the future drug development.

# Materials and methods

## Key resources table

| Reagent type (species) or resource | Designation | Source or reference | Identifiers | Additional information |
|---|---|---|---|---|
| Gene (human) | pOPINS, EKAR2G_design1_mTFP_wt_Venus_wt | UniProt | Q15691, MAPRE1 | EBH domain of EB1, expression construct |
| Strain, strain background (*Escherichia coli*) | BL21(DE3) | NEB | C2530H | Chemically competent cells |
| Cell line (human) | HeLa | AATC | PCS-201–012, RRID:CVCL_0030 | Mycoplasma negative |
| Transfected construct (human) | EKAR2G_design1_mTFP_wt_Venus_wt | Addgene | Plasmid #39813, RRID:Addgene_39813 | FRET vector |
| Transfected construct (human: MAPRE1) | mCherry-EB1-8 | Addgene | Plasmid #55035, RRID:Addgene_55035 | C-terminal mCherry tag |
| Recombinant DNA reagent | pOPINS (plasmid) | Addgene | Plasmid #41115, RRID:Addgene_41115 | SUMO-tag expression vector |
| Peptide, recombinant protein | EBH fragment | This paper | | Protein preparation is described in Materials and methods. Plasmid for the expression available from the corresponding author. |

*Continued on next page*

*Continued*

| Reagent type (species) or resource | Designation | Source or reference | Identifiers | Additional information |
|---|---|---|---|---|
| Peptide, recombinant protein | SxIP peptides | ChinaPeptides (Shanghai) | | Set of synthetic peptides |
| Commercial assay or kit | In-Fusion HD Cloning | Clontech | Clontech:639647, RRID:SCR_004423 | |
| Chemical compound, drug | CBR-5884 | Sigma-Aldrich | SML1656, RRID:SCR_008988 | |
| Chemical compound, drug | Lipofectamine 3000 | Invitrogen | #L3000015 | Transfection reagent |
| Software, algorithm | SPSS | SPSS | RRID:SCR_002865 | |
| Software, algorithm | Microcal PEAQ-ITC Software, v. 1.41 | Malvern | RRID:SCR_023795 | ITC data analysis |
| Software, algorithm | TopSpin 4.3 | Bruker | RRID:SCR_014227 | NMR data processing |
| Software, algorithm | CCPNmr Analysis v2.4 | CCPN | RRID:SCR_016984 | NMR data analysis software |
| Software, algorithm | Relax 4.1.1 | PMID:18085411 | | NMR relaxation analysis |
| Software, algorithm | ChemEx 2018.10.2 | https://github.com/gbouvignies/chemex; *Bouvignies, 2025* | | CEST data analysis |
| Other | DAPI stain | Invitrogen | D1306 | Commercial nuclear stain (1 µg/mL) |

## Peptides and protein preparation

Human EB1 (Uniprot code Q15691), residues 191–260 and 191–252, was purified as previously described (*Almeida et al., 2017*).

Synthetic MACF peptides with purity >95% were purchased from ChinaPeptides (Shanghai). Peptide amino acid sequences are: 4MACF (SKIP), 6MACF (SKIPTP), 9MACF (SKIPTPQRK), 11MACF (KPSKIPTPQRK), 11MACF-VLL (KPSTAKSKIPVLL), 11MACF-LLL (KPSTAKSKIPLLL), 11MACF-VLLRK (KPSTAKSKIPVLLRK), and 13CK5P2 (KKSRLPRILIKRSR). Peptides were dissolved with Milli-Q ultrapure water to make 10–20 mM stock solutions; pH was adjusted to 7 with NaOH.

## NMR spectroscopy

NMR spectra were collected on Bruker Avance III 600 and Neo 800 MHz spectrometers equipped with CryoProbes. Spectra were processed with TopSpin 4.3 (Bruker) and analysed using CCPNmr Analysis v2.4 (*Vranken et al., 2005*). Experiments were performed at 298 K in 20 mM phosphate pH 6.5, 50 mM NaCl, 0.5 mM TCEP, 0.02% (wt/vol) $NaN_3$.

To achieve full saturation of the complex, data for structure calculations was collected with a slight excess of ligand and 10:11 EB1/11MACF ratio. The backbone resonances were assigned using triple resonance experiments (HNCO, HN(CA)CO, HNCA, HNCACB, and CBCACONH) measured for $^{13}C$-$^{15}N$ labelled EB1 using standard assignment protocols in CCPN analysis.

Side chain resonance assignments were obtained using HBHA(CO)NH, H(C)CH-TOCSY, and (H)CCH-TOCSY experiments. Aromatic side chains were assigned using 2D-NOESY and $^1H$-$^{13}C$-resolved-NOESY-HSQC. The resonances of the ligands were assigned using $^{13}C$,$^{15}N$-filtered 2D TOCSY and NOESY experiments. The structures were calculated using ARIA 2.2 integrated with CCPNmr analysis (*Vranken et al., 2005*), as fully described previously (*Almeida et al., 2017*). Statistics of the structure determination are presented in *Appendix 1—table 1*.

Standard Bruker $^1H$,$^{15}N$-HSQC sequence hsqcfpf3gpphwg with soft-pulse watergate water suppression and flip-back pulse was used for the NMR lineshape analysis. A series of $^1H$,$^{15}N$-HSQC spectra were collected for the $^{15}N$-labelled EBH domain with variable concentrations of the peptide. The range of the peptide concentrations has been selected to ensure maximum saturation at the highest concentration and even distribution of the saturation states throughout the titration. The concentrations used are listed in the supplementary material (see *Figure 1—figure supplements 1 and 3*). The titration HSQC data were analysed in TITAN software (v1.6) (*Waudby et al., 2016*). For the EBH-ΔC/11MACF, the 1:1 binding model was used because chemical shift changes were linear

and no allosteric effect from binding to the two different binding sites in the dimer was detected. For EBH-/11MACF and EBH-ΔC/11MACF-VLL, we observed non-linear changes of chemical shifts and multiple HSQC peaks corresponding to the non-symmetrical complex where only one binding site in the dimer is occupied. To account for this, we used the dimer binding model implemented in TITAN. For each residue with a significant chemical shift perturbation that can be followed throughout the titration, lineshapes for the free state and the final titration point were initially fitted to evaluate the parameters of the free state and the complex. Then, the titration series was fitted separately for each residue to obtain residue-specific exchange parameters. Finally, the lineshapes were fitted for all selected residues simultaneously to evaluate the global exchange rate.

The $^{15}N$ CEST experiments (*Vallurupalli et al., 2012*) were conducted for samples containing 0.75 mM $^{15}N$ EB1 and 2.5% (molar) of the MACF peptide at a $^1H$ frequency of 800 MHz and 298 K. CEST profiles were measured at $^{15}N$ B1 field strength of 12.5, 25, and 50 Hz applied during a constant period of 400 ms using the standard Bruker pulse sequence. For the residues with detected CEST NMR exchange, data at all B1 values were fitted separately for each residue and simultaneously for all the residues using the ChemEx software (https://github.com/gbouvignies/chemex) as described previously (*Vallurupalli et al., 2012*).

The $^{15}N$ relaxation rates $R_1$, $R_2$, and $\{^1H\}$-$^{15}N$ heteronuclear Overhauser effects (nOes) were measured at $^1H$ frequencies of 600 and 800 MHz and 298 K using HSQC-based pulse sequences using standard Bruker pulse sequences. EBH concentration on the samples was 0.5 mM; 3 mM MACF peptide was used to ensure full saturation of the complex. To obtain the dynamic parameters, the data were analysed in the Relax 4.1.3 software (*d'Auvergne and Gooley, 2008*) using standard protocols.

## Isothermal titration calorimetry

ITC experiments were performed using the MicroCal ITC200 and Malvern MicroCal Automated PEAQ-ITC instruments. Both protein and peptides were dialysed into 50 mM phosphate (pH 6.5), 50 mM NaCl, 0.5 mM TCEP, and 0.02% NaN₃ before use. To test the salt dependence of $K_D$, the salt concentration in the buffer was increased to 150 mM NaCl. The sample cell and syringe were filled with 20–40 µM EB1 EBH domain and 200–600 µM MACF peptide, respectively. 1.5 µL of MACF peptides were injected into the sample cell for a total of 25 injections. All experiments were performed in triplicate and at 25°C. The data were integrated and fitted into the single-site binding model using the Malvern PEAQ-ITC Control software v1.41. The experiments were performed in triplicates to evaluate standard deviation. Data were analysed using Microcal PEAQ-ITC software v1.41.

## Cell culture

HeLa cells were cultured in Dulbecco's Modified Eagle Serum substituted with 10% fetal bovine serum and 1% penicillin/streptavidin. Cells were maintained at 37°C and 5% CO₂. HeLa cells were obtained from the ATCC (CCL-2), authenticated by provider using STR profiling. Negative for mycoplasma.

## MACF peptide constructs and transfection

Peptide sequences WT (GSRPSTAKPSKIPTPQRK), VLLRK (GSRPSTAKPSKIPVLLRKRK), and dimeric LZ-VLLRK peptide containing GCN4p1 leucine zipper (RMKQLEDKVEELLSKNYHLENEVARLKKLVGE RGSRPSTAKPSKIPVLLRK) were inserted into a EKAR2G_design1_mTFP_wt_Venus_wt vector courtesy of Oliver Pertz (Addgene plasmid #39813; http://n2t.net/addgene:39813; RRID:Addgene_39813). The peptides were inserted at the mTFP C-terminus between the BspEI and EcoRV restriction sites of the vector. The design followed *Honnappa et al., 2009*, to allow for the direct comparison with the published results. The peptides were separated from mTFP by the GS sequence of the vector and contained another GS at the N-terminus, creating a highly flexible GSGS linker that minimises potential effect of mTFP on the interaction.

Peptide constructs were co-transfected into HeLa cells with an mCherry-EB1 construct using Lipofectamine 3000 reagent following the manufacturer's instructions (Invitrogen #L3000015). mCherry-EB1-8 was a gift from Michael Davidson (Addgene plasmid #55035, http://n2t.net/addgene:55035; RRID:Addgene_55035). Imaging was performed 24 hr after transfection.

## Live-cell imaging

Live-cell images were obtained using the Marianas spinning disk confocal microscope system (Intelligent Imaging Innovations, Inc) and Slidebook2022 (reference) capture software. Total internal reflection microscopy (TIRFM) was employed using 100×1.49 NA lens, 488 (FF01-525/30-25) and 561 (FF01-617/73-25) lasers with images captured using a Hamamatsu C11440 camera.

## Acknowledgements

TA was funded by Liverpool University PhD studentship. EH was funded by BBSRC DTP NLD studentship. The NMR spectra were measured at the LIV-SRF High-Field NMR Facility, University of Liverpool. The images were collected at the LIV-SRF Centre for Cell Imaging (CCI) Facility, University of Liverpool.

## Additional information

### Funding

| Funder | Grant reference number | Author |
|---|---|---|
| Biotechnology and Biological Sciences Research Council | DTP | Eleanor Hargreaves |
| Liverpool University PhD studentship | | Teresa Almeida |

The funders had no role in study design, data collection and interpretation, or the decision to submit the work for publication.

### Author contributions

Teresa Almeida, Investigation, Writing – original draft; Eleanor Hargreaves, Investigation, Visualization, Writing – original draft, Writing – review and editing; Tobias Zech, Conceptualization, Resources, Supervision, Writing – original draft; Igor Barsukov, Conceptualization, Formal analysis, Supervision, Methodology, Writing – original draft, Writing – review and editing

### Author ORCIDs

Teresa Almeida ⓘ https://orcid.org/0000-0001-9308-8678
Eleanor Hargreaves ⓘ https://orcid.org/0000-0002-9287-8480
Igor Barsukov ⓘ https://orcid.org/0000-0003-4406-9803

Reviewer #1 (Public review): https://doi.org/10.7554/eLife.98063.3.sa1
Reviewer #2 (Public review): https://doi.org/10.7554/eLife.98063.3.sa2
Author response https://doi.org/10.7554/eLife.98063.3.sa3

## Additional files

### Supplementary files

MDAR checklist

### Data availability

The coordinates for the solution structure of EBH/MACF complex are deposited in the PDB, accession code 7OLG (https://doi.org/10.2210/pdb7olg/pdb). The chemical shifts are deposited in the BMRB, accession code 34629. The relaxation data are deposited to BMRB, access codes 53187 (free EBH) and 53188 (EBH/11MACF complex).

The following datasets were generated:

| Author(s) | Year | Dataset title | Dataset URL | Database and Identifier |
|---|---|---|---|---|
| Almeida TB, Barsukov IL | 2021 | EB1 bound to MACF peptide | https://doi.org/10.2210/pdb7olg/pdb | Worldwide Protein Data Bank, 10.2210/pdb7olg/pdb |
| Almeida TB, Barsukov IL | 2022 | EB1 bound to MACF peptide | https://bmrb.io/data_library/summary/index.php?bmrbId=34629 | BMRB, 34629 |
| Barsukov I, Hargreaves E | 2025 | EB1 EBH(191-260) relaxation data | https://bmrb.io/data_library/summary/index.php?bmrbId=53187 | BMRB, 53187 |
| Barsukov I, Hargreaves E | 2025 | EB1 EBH(191-260) bound to MACF peptide, relaxation data | https://bmrb.io/data_library/summary/index.php?bmrbId=53188 | BMRB, 53188 |

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

## Appendix 1

### TITAN models for fitting EBH titration data

In general, the dimeric structure of EBH and the location of the binding site at the interface between the monomers requires the use of the dimer binding model with three different states: free EBH, EBH bound to one peptide, and the full complex with two peptides bound. In both the free and the final complex, the chemical shifts of both subunits are identical due to the symmetry of the EBH dimer. However, binding to only one site may have an indirect effect on the chemical shifts of the other site and the appearance of two additional signals at the intermediate points of the titration: one corresponding to the empty and one to the filled site that, which differ from the shifts of the free and fully bound states. The indirect effects on the chemical shifts are usually small, leading to the cross-peaks of the empty site being close to the free state, and the occupied site to the peaks of the fully bound state. These additional cross-peaks can only be observed if the exchange rate is sufficiently slow. In the case of the fast exchange, the presence of the intermediate state can be detected as a complex, non-linear pattern of the chemical shifts (*Waudby et al., 2016*). In such cases, a four-state model is required for the fitting. However, if the indirect effect on the chemical shifts is small and the exchange is fast, the binding to each site can be treated as independent, and the changes in the spectra can be fitted with a two-state binding model.

We observed different exchange regimes for the interactions of EBH and EBH-ΔC with different MACF peptides, in agreement with the dissociation constants (*Table 1*). The interaction of the 11MACF with EBH-ΔC (ITC $K_D$ 41.5 ± 8.8 µM) showed gradual chemical shift changes with a significant peak broadening for the largest chemical shift differences, corresponding to the fast-intermediate exchange (*Figure 4—figure supplement 2A*). The majority of the peak trajectories followed a straight line, in agreement with small indirect binding effect and applicability of the two-site binding model. The TITAN analysis with the single site model (independent binding to the two binding sites of the dimer) gave $K_D$ = 26.6 ± 0.051 µM that agreed well with the ITC $K_D$ and $k_{off}$ = 1900 ± 18 s⁻¹, corresponding to the fast exchange. The mutant 11MACF-VLL peptide showed stronger binding by ITC (ITC $K_D$ 18.7 ± 3.1 µM). In agreement with this, we observed more pronounced broadening during the NMR titration (*Figure 4—figure supplement 3B*). TITAN analysis of the data gave $K_D$ = 25.62 ± 0.82 µM that agreed well with the ITC $K_D$ and $k_{off}$ = 1541 ± 79 s⁻¹, that was close to the off-rate of the WT peptide. This demonstrated that the affinity increase was caused mainly by the on-rate increase.

For the significantly stronger interaction of the 11MACF with EBH (ITC $K_D$ 3.5 ± 1.0 µM), we observed a combination of intermediate exchange for smaller chemical shift changes and slow exchange for the largest changes in signals of the residues in the binding pocket, particularly in the C-terminal region that becomes immobilised in the complex (*Figure 4—figure supplement 2B*). In a number of cases, the peak trajectories were not linear, indicating an indirect effect on chemical shifts in one site from the binding of the peptide to the other site. At the lower temperature of 15°C, additional peaks were observed for the intermediate binding state. These signals correspond to the non-symmetrical EBH complex where one binding site is occupied, and the other binding site is empty. Although the two binding sites are located at the opposite sides of the EBH dimer and do not overlap, the peptide binding to one site causes small changes in the EBH structure propagated to the unoccupied site. This, in turn, leads to the small chemical shift changes in the empty site. In a similar way, the binding of the second peptide causes small structural changes in the occupied site and, thus, additional chemical shift changes. As a result, in the non-symmetrical complex with one peptide bound, the signals of free site are different from the signals of the free EBH, and the signals of the bound site are different from the signals of the symmetrical complex with the two peptides bound. This leads to the appearance of additional signals in the intermediate bound state when the exchange is slow, or to non-linear signal behaviour under sufficiently fast exchange conditions. The additional signals confirm the binding to the EBH dimer. The small value of the chemical shift changes induced by the indirect effect shows that the structural changes are very small.

In the case of non-linear chemical shift changes or appearance of new signals, the single-side binding model cannot reproduce the spectral variations and the dimer four-state binding model with additional parameters has to be used for the fitting. The dimer binding model in TITAN includes cooperativity factors $\alpha = \log(K_{D,2}/K_D)$ and $\beta = \log(k_{off,2}/k_{off})$, where $K_{D,2}$ and $k_{off,2}$ are the

dissociation constant and off-rate of the second binding, respectively. The fitting of the titration with all parameters showed close to zero values for both cooperativity parameters. We, therefore, fixed their values at zero during the fit to simplify the model. The resulting values of $K_D = 4.9 \pm 0.1$ µM and $k_{off} = 130.2 \pm 2.1\,\mathrm{s}^{-1}$ were close to the values obtained with the cooperativity parameters ($K_D = 4.8 \pm 0.1$ µM and $k_{off} = 130 \pm 0.7\,\mathrm{s}^{-1}$).

For the stronger interaction between 11MACF-VLL and EBH (ITC Kd 80 nM), many signals showed slow exchange and clear presence of signals from the intermediate binding state (*Figure 4—figure supplement 2C*), which required the use of the dimer binding model. In most cases, one of the intermediate state signals was located close to the free, and the other close to the fully bound-state signal, in agreement with a much smaller indirect effect. However, for the residues in the loop between the short and the long helix Gly230, Asn231, Gly233, and Asn235, signals of the intermediate state were significantly more displaced from the signals of the free and fully bonded forms (*Figure 4—figure supplement 2C*). These residues are located far from the binding site and do not make direct contact with the peptide. The effect on the chemical shifts of these residues indicates some structural rearrangement of the relative positions of the helices induced by the peptide binding to each of the sites.

The fitting of the 11MACF-VLL titration with all four parameters resulted in $K_D = 66 \pm 2\,\mathrm{nM}$, $k_{off} = 15 \pm 0.2\,\mathrm{s}^{-1}$, $\alpha = 0.5$ and $\beta = 0$, corresponding to a weak negative cooperativity with an ~3-fold increase of the $K_D$ for the second binding without a change in the dissociation rate. The dissociation constant for the first binding event agreed very well with the ITC dissociation constant of 80 ± 2 nM, supporting the NMR analysis. The negative cooperativity may reflect the small structural rearrangement detected from the chemical shift changes. When the cooperativity parameters were set to zero during the fit, corresponding to the two-state binding model used for ITC, the $K_D$ value increased to 200 ± 4 nM, while the dissociation rate remained unchanged. The lack of the systematic deviations between the experimental ITC data and the single-site analysis model shows that the difference between the first and the second site binding, if it exists, is too small to be detected by ITC, in line with the weak cooperativity observed by NMR. We therefore concluded that while NMR gives indication of the weak negative cooperativity, the differences in the binding affinities for the first and second ligand binding events are too small for a reliable interpretation, and the single-site binding model should be used to avoid overfitting and overinterpretation.

## Exchange approximation for low population of the intermediate state

From $K_D'$ and $k_{off}'$ values for the EBH/11MACF interaction, we can estimate $K_D^L = 0.08 - 0.07$, showing that only ~8% of the bound population is in the intermediate dock state. Since the intensities of the NMR signals corresponding to the intermediate state are proportional to the population of this state, its contribution to the spectra is negligibly small compared to the fully bound state and can be neglected. Under this condition, the measured line shape changes and CEST saturation transfer are only associated with the exchange between the free and the fully bound forms. These rates can be calculated as the overall fluxes between the states using the steady-state conditions as (*Shoup and Szabo, 1982*):

$$k_{on}' = k_{on}^D \frac{k_{on}^L}{k_{off}^D + k_{on}^L} \tag{A1}$$

$$k_{off}' = k_{off}^D \frac{k_{off}^L}{k_{off}^D + k_{on}^L} \tag{A2}$$

The effective dissociation constant between these states is then:

$$K_D' = K_D^D * K_D^L \tag{A3}$$

which agrees with *Equation 1* when $K_D^D \ll 1$.

*Equation 3* can be rearranged:

$$k'_{on} = k^D_{on} \frac{1}{\frac{k^D_{off}}{k^L_{on}} + 1}$$

(A4)

Since

$$\frac{1}{\frac{k^D_{off}}{k^L_{on}} + 1} \leq 1$$

(A5)

then

$$k'_{on} \leq k^D_{on}$$

(A6)

From this equation, the additional exchange step caused by the conformation change of the complex should generally lead to a reduction in the on-rate for the full transition to the complex. In the case of a very fast transfer from the intermediate to the fully bound state compared to the ligand dissociation from the transition state ($k^D_{off} \ll k^L_{on}$), the initial dock step becomes rate-limited, with overall on-rate equal to the on-rate of the dock step.

From *Equations 1 and 2* of the main text,

$$\frac{K'_D}{k'_{off}} = \frac{K^D_D}{k^D_{off}}$$

(A7)

and

$$k'_{on} = k^D_{on}$$

(A8)

So, the equality of the on-rates also holds under fast exchange in the second lock step even if the population of the intermediate state is not negligibly small.

When the population of the intermediate state is comparable to the population of the fully bound state and the exchange in the second lock step is intermediate or slow, the binding cannot be approximated by a two-site model and the intermediate step should be considered explicitly. Under these conditions, NMR signals of the fully titrated protein will have an exchange contribution or additional signals corresponding to intermediate states will be observed. This will allow direct evaluation of the exchange parameters of the second lock step from the NMR analysis.

**Appendix 1—table 1.** NMR restraints and structure statistics for the NMR structure of the EBH/11MACF complex.

| Total restraints used | |
| --- | --- |
| NOE restraints* | |
| All | 3863 |
| Protein-ligand | 298 |
| Intermonomer | 924 |
| Intrapeptide | 209 |
| Intraresidue | 1133 |
| Sequential ($|i - j|$=1) | 968 |
| Medium ($1 < |i - j| \leq 4$) | 1311 |
| Long range ($|i - j| > 4$) | 177 |
| Dihedral | |

*Appendix 1—table 1 Continued on next page*

*Appendix 1—table 1 Continued*

**Total restraints used**

| | |
|---|---|
| Φ angles | 65 |
| φ angles | 65 |
| Hydrogen bonds | 90 |
| Structure statistics | |
| Violations | |
| Distance (>0.5 Å) | 41 |
| Dihedral angle (>5°) | 7 |
| Energies (cal/mol) | |
| Overall | −2179 (±208) |
| Bond | 119 (±9) |
| Angle | 479 (±21) |
| Improper | 242 (±29) |
| Dihedral | 868 (±14) |
| Van der Waals | −48 (±28) |
| Electrostatic | −5903 (±98) |
| NOE | 1906 (±119) |
| Geometry – average values | |
| Bond | $7.40 \times 10^{-3}$ (±$5.7 \times 10^{-4}$) |
| Angle | 0.91 (±$9.96 \times 10^{-2}$) |
| Improper | 2.50 (±0.30) |
| Dihedral | 41.56 (±0.24) |
| Van der Waals | 428.93 (±83.98) |
| Average pairwise RMSD (Å)[†] | |
| Heavy atoms | 2.41 (±1.25) |
| Heavy atoms – helical region | 1.51 (±0.99) |
| Backbone | 2.05 (±1.33) |
| Backbone – helical region | 1.08 (±0.93) |
| Ramachandran statistics (%) [‡] | |
| Most favoured regions | 87.0 (99.5) |
| Additional allowed regions | 10.8 (0.3) |
| Generously allowed regions | 1.0 (0.2) |
| Disallowed regions | 1.1 (0) |

*Number in brackets corresponds to the restraints assigned manually.

[†]Helical region corresponds to residues: Glu192-Glu230 and Pro237-Tyr247.

[‡]Values within brackets correspond to residues Glu192-Glu230 and Pro237-Tyr247 (helical region).

