## [Editor Report · eLife Assessment]

This study provides **important** insights into how the EBH domain of microtubule end-binding protein 1 (EB1) interacts with SxIP peptides derived from the MACF plus-end tracking protein. The revised manuscript includes **convincing** ITC and NMR experiments that clarify the role of flanking residues and address the influence of dimerization and cooperativity on binding. While some mechanistic aspects remain difficult to resolve experimentally, the data and analysis now more clearly justify the proposed "dock-and-lock" model and its interpretive value. This work will be of interest to structural biologists and biophysicists studying microtubule-associated protein interactions.

---

## [Referee Report · Reviewer #1 (Public review)]

Summary:

In this article, Almeida and colleagues use a combination of NMR and ITC to study the interaction of the EBH domain of microtubule end-binding protein 1 (EB1) with SxIP peptides derived from the MACF plus-end tracking protein. EBH forms a dimer and in isolation has previously been shown to have a disordered C-terminal tail. Here, the authors use NMR to determine a solution structure of the EBH dimer bound to 11-mer SxIP peptides derived from MACF, and observe that the disordered C-terminal of EBH is recruited by residues C-terminal to the SxIP motif to fold into the final complex. By comparison of binding in different length peptides, and of EBH lacking the C-terminal tail, they show that these additional contacts increase binding affinity by an order of magnitude, greatly stabilising the interaction, in a binding mode they term 'dock-and-lock'.

The authors also use their new structural knowledge to design peptides with higher affinities, and show in a cell model that these can be weakly recruited to microtubule ends - although a dimeric construct is necessary for efficient recruitment. Ultimately, by demonstrating the feasibility of targeting these proteins, this work points towards the possibility of designing small-molecules to block the interactions.

---

## [Referee Report · Reviewer #2 (Public review)]

Summary:

The C-terminal region of EB1 is responsible for protein-protein interactions, thereby recruiting the binding partners of EB1 to microtubules; the coiled-coil region (EBH) and the acidic tail are critical for their binding partners. The authors demonstrated by using NMR that the binding mode of EBH with the SxIP motif, which is a two-step process termed "dock-and-lock". The ITC analysis supports the results obtained from NMR. The initial version of the manuscript contained ambiguities on the ITC data; however, the results of the revised manuscript are convincing and support the two-step binding model.

Strength:

The authors propose a novel model of "dock-and-lock" by using multiple methods of NMR, ITC and cell biology.

---

## [Author Response]

The following is the authors’ response to the original reviews

We would like to express our sincere gratitude to the reviewers for their thorough analysis of the manuscript and their extremely helpful comments. We have taken all the suggestions into consideration and conducted a range of additional experiments to address the points raised. We have also extensively revised the manuscript to clarify descriptions, correct inaccuracies and remove inconsistencies. We have modified the figures for clarity and content.

Overall, we expanded the description of the EBH structure to emphasise its dimeric nature and the impact of the two binding sites on interpreting the binding data, including cooperativity. Using ITC, we tested the effect of the pre-SxIP residues on the binding affinity with additional peptides. We found that these residues had a significant effect, albeit much smaller than that of the post-SxIP residues. We analysed the binding of the 11MACF-VLL mutant with EBH-ΔC and evaluated the exchange rates. In agreement with our model, we found that the EBH affinity for the SxIP peptide from CK5P2 (KKSRLPRILIKRSR), which has a C-terminal sequence similar to that of the 11MACF-VLLRK mutant, is 21nM, which is similar to the affinity of the mutant itself. This demonstrates the significant variation in affinity observed among natural SxIP ligands, as predicted by our study. Our responses to the specific points raised by the reviewers are provided below.

**Reviewer #1 (Public Review):**
There is no direct experimental evidence for independent dock and lock steps. The model is certainly plausible given their structural data, but all titration and CEST measurements are fully consistent with a simple one-step binding mechanism. Indeed, it is acknowledged that the results for the VLL peptide are not consistent with the predictions of this model, as affinity and dissociation rates do not co-vary. The model may still be a helpful way to interpret and discuss their results, and may indeed be the correct mechanism, but this has not yet been proven.

Unfortunately, it is not possible to obtain direct experimental evidence because the folding of the C-terminus is too fast to influence the NMR parameters. However, as the reviewer pointed out, our structural data support the two-step model, since folding of the C-terminus is only possible once the ligand containing the post-SxIP residues has bound. By adopting a mechanistically supported model, we can analyse the contributions to binding and relate them to the structural characteristics of the complex. This provides a clearer insight into the roles of the various regions in the interaction and allows to modify them rationally to enhance the ligand affinity.

In the revised version, we restate the equations in terms of comparing the on-rates. This provides a clearer view of the effect of the additional stage, which cannot increase the overall on-rate since the two stages are sequential. If the forward rate of the second stage is comparable to or slower than the off-rate of the first stage, the overall on-rate decreases. Conversely, if the forward rate is much faster, the overall on-rate remains unchanged. For the wild-type 11MACF peptide, we observed that the presence of the EBH C-terminus does not affect the on-rate of binding, which is in perfect agreement with the two-step model and indicates that the C-terminus folds very quickly.

Additionally, we evaluated the binding of the 11MACF-VLL mutant to EBH-ΔC and observed a twofold decrease in Kd compared to WT 11MAC, primarily due to an increase in the on-rate. Interestingly, this rate is approximately twice as low as the overall on-rate for EBH/11MACF-VLL binding, contradicting the sequential two-step model. This suggests a more complex binding process where binding is accelerated by additional hydrophobic interactions with the unfolded C-terminus. However, given the difficulty of quantifying very slow exchange rates, it is more likely that the discrepancy is due to the accuracy of the rate measurements. Therefore, the model allows the rational analysis of changes in binding parameters due to mutations.

There is little discussion of the fact that binding occurs to EBH dimers - either in terms of the functional significance of this or in the acquisition and analysis of their data. There is no discussion of cooperation in binding (or its absence), either in the analysis of NMR titrations or in ITC measurements. Complete ITC fit results have not been reported so it is not possible to evaluate this for oneself.

We added information about the dimer to the introduction, emphasising its role in enhancing interaction with microtubules (MTs) and its structural role in SxIP binding. The ITC data do not exhibit any biphasic behaviour and can be fitted to a single-site model with 1:1 stoichiometry relative to the EB1c monomer. This corresponds to two independent binding sites in the dimer. We have added the stoichiometry to Table 1 and the description. The NMR titration data for the 11MACF and 11MACF-VLL interactions were fitted to the TITAN dimer model, which includes cooperativity parameters. For WT 11MACF, both cooperativity parameters were zero, corresponding to independent binding sites in the ITC model. For 11MACF-VLL, the fitting suggests weak negative cooperativity, with a ~3-fold increase in Kd for binding to the second site and no change in the off-rate. This difference in Kd is likely to be too small to induce a biphasic shape to the ITC curve. As the cooperativity effect on the NMR spectra is small and absent in the ITC, we used the independent sites model for data analysis, as there is insufficient justification for introducing extra parameters into the model. Crucially, fitting to this model did not alter the off-rate value obtained by NMR or affect the conclusions. We added a description of cooperativity to the results and discussion.

Three peptides are used to examine the role of C-terminal residues in SxIP motifs: 4-MACF (SKIP), 6-MACF (SKIPTP), and 11-MACF (KPSKIPTPQRK). The 11-mer demonstrates the strongest binding, but this has added residues to the N-terminal as well. It has also introduced charges at both termini, further complicating the interpretation of changes in binding affinities. Given this, I do not believe the authors can reasonably attribute increased affinities solely to post-SxIP residues.

We tested the 9MACF peptide SKIPTPQRK, which has the same N-terminus as the 4- and 6-MACF peptides, and found that its binding affinity is ~10-fold weaker than that of 11MACF. This demonstrates the contribution of both the pre- and post-SxIP residues. This is likely due to electrostatic interactions between the positively charged N-terminus and the negatively charged EBH surface, similar to those involving the positive charges at the peptide C-terminus. Although significant, the contribution of the N-terminal peptide region is approximately one order of magnitude lower than that of the post-SxIP residues, meaning the post-SxIP region is the main affinity modulator. We have added the binding data on 9MACF and a discussion of the contributions to the manuscript.

Experimental uncertainties are, with exceptions, not reported.

Uncertainties added to the number in Table 1 and the text. Information on how uncertainties were calculated added to Table 1.

**Reviewer #1 (Recommendations For The Authors):**
(1) Have you tested the binding of the WT dimer in your cell model?

We haven’t tested the WT dimer because it has already been reported in the 2009 Cell paper by Honappa et al. In the cell experiments, our main focus was on recruiting the high-affinity mutant to MTs. The low level of recruitment, despite the mutant's high affinity, highlights the importance of dimerisation or additional contributions to binding.

(2) Please deposit all NMR dynamics measurements (relaxation rates and derived model-free parameters) alongside structural data in the BMRB.

The relaxation data have been submitted to BMRB, IDs 53187 and 53188

(3) Please report complete fitting results, e.g. for ITC, including stoichiometries. Clarify what this means for binding to a dimer, and if there is any evidence of cooperativity. Figure 3C, right hand panel, shows an unusual stoichiometry, can the authors comment on this?

We have added more information on stoichiometry and cooperativity; please refer to our response to the above comment for details. We repeated the titration for the VLLRK mutant using fresh peptide stock. As expected, the stoichiometry was close to 1:1 relative to the EB1c monomer. The new data are now included in the table and figure.

(4) Please report uncertainties for all measurements of Kd, koff, kon, ∆G, ∆H, ∆S, and explain whether these are determined from statistical analysis, technical or biological repeats (and where reported, clarify between standard deviation/standard error). Please also be aware of standard guidelines for reporting significant figures for data with uncertainties, as these have not been followed in Table 1.

Uncertainties added to the number in Table 1 and the text. Information on how uncertainties were calculated added to Table 1.

(5) The construct design for the cell model is unclear - given the importance of flanking residues, please report and discuss how the sequences are attached to venus: which termini is attached, and what is the linker composition?

We cloned the peptides at the C-terminus of mTFP, after the GS linker of the vector. The peptide itself contains a GS sequence at the N-terminus, creating a highly flexible GSGS linker that separates the SxIP region from mTFP and minimises the potential effect of mTFP on binding. We followed the design of Honappa et al. to enable direct comparison with the published results. We have added this information to the 'Methods' section..

(6) Which HSQC pulse sequence was used for 2D lineshape analysis? The authors mention non-linear chemical shift changes, presumably associated with the dimer interface - this would be useful to expand upon and clarify.

For the lineshape analysis, we used the standard Bruker sequence hsqcfpf3gpphwg with soft-pulse watergate water suppression and flip-back. This sequence is included in the TITAN model. We added the description of the non-linear chemical shift changes and connection of these changes to the allosteric effect of the binding to the supplementary information describing details of the lineshape analysis.

(7) Figure 1A could usefully highlight the dimer interface in the surface representation also.

We believe that including the interface would make the figure too complicated. The dimer configuration is shown in different colours for the two subunits, clearly demonstrating their involvement in forming the binding site.

(8) Figures 1C and 1D could usefully show a secondary structure schematic to assist the reader. The x-axis in these figures is not linear and this should be corrected. The calculation of combined chemical shift perturbations should be described.

Thank you for the helpful suggestion. We changed the scale of the figures and added the diagram of the secondary structure.

(9) Units are missing from many figure axes.

We added missing units to the axes. Thank you for highlighting this.

(10) What peptide concentrations are used in Figure 1C? Presumably, these should be reported at saturation for this to be a fair comparison, this should be clarified.

The protein concentration was 50 µM. Peptides 4MACF and 6MACF were added at a 100-fold molar excess and peptide 11MACF was added at a 4-fold excess. Saturation was achieved for 11MACF. This was impossible for the short peptides due to their mM affinity. This information has been added to the figure legend. The figure's main aim is to illustrate the differences in the chemical shift perturbation profiles, which can be achieved even if full saturation is not attained. Although the absolute value of the chemical shifts is proportional to the degree of saturation, the distribution of the largest chemical shift changes is independent of this degree. Therefore, we can draw conclusions about the distribution of changes by comparing under non-saturation conditions.

(11) The presentation of raw peak intensities in Figure 1D shows primarily the flexibility of the C-terminal region associated with high intensities. Beyond this, when comparing the binding of peptides it would be much more informative to show relative peak intensities. Residues around 210-225 appear to show strong broadening in the presence of peptide, but this is masked by the low initial intensity. Can the authors clarify and discuss this? Also, what peptide concentrations were used for this comparison? For a fair comparison, it should be close to saturation - particularly to exclude exchange broadening contributions.

The protein concentration was 50 µM. 6MACF and 6MACF peptides were added at a 100-fold excess and 11MACF at a 4-fold excess. Saturation was achieved for 11MACF. This was impossible to achieve for the short peptide due to its mM affinity. This information has been added to the figure legend. Upon checking the data, we found a small systematic offset in the coiled-coil region of some of the complexes, as the integral intensity had been used in the initial plot. While this does not change the conclusion regarding the high dynamics of the C-terminus, it does create an inaccurate perception of the relative intensities of the folded regions in the different complexes, as noted by the reviewer. We have now plotted the amplitudes at the maximum of the peaks, which do not exhibit any systematic offset as they are much less susceptible to baseline distortions. We are grateful to the reviewer for highlighting this apparent discrepancy.

(12) Figure 2 - the scale for S2 order parameters appears to be backwards, given the caption, but its range should be indicated. Similarly, the range of values for Rex should also be indicated. These data should also be tabulated/plotted in supporting information.

We have corrected the figure legend and added S2 and Rex plots to the supplementary material. The figure aims to highlight regions of increased mobility, while the plots provide full quantitative information on the values. We thank the reviewer for pointing out the error in the figure legend and for the suggestions regarding the plots.

(13) The scale in Figure 3B is illegible. Indeed, the whole structure is quite small and could usefully be expanded.

We increased the size of the structure panels and added a scale.

(14) Figure 4 does not show a decrease in exchange rates, as per the caption - no comparison of exchange rates is shown, only thermodynamic information in panel E. Panel C shows CEST measurements, but it is not clear what system this is for - please clarify, and consider showing the comparable data for the ∆C construct for comparison.

We have amended the figure legend to clarify that the figure shows binding parameters. We added information about the CEST profiles for the EBH/11MACF interaction to the figure legend (Figure 4C). Exchange with the ∆C construct is too fast for CEST measurements. We used lineshape analysis to evaluate the exchange rates for this construct.

(15) The schematics shown in Figure 4D, and elsewhere, are really quite difficult to understand. They may pose additional challenges to colourblind readers. Please consider ways that this could be clarified.

We simplified the colour scheme in the model to make the colours easier to see and to highlight SxIP and non-SxIP regions. We believe that this improved the clarity of the figure.

(16) Figures S1D/E - the x-axes are unclear and units are missing from the y-axes.

We re-labelled the axes to clarify the scale and units. Thank you for pointing this.

**Reviewer #2 (Public Review):**
The C-terminal tail of EB1, which is adjacent to EBH and is not analyzed in this study, is highly acidic and plays an important role in protein interactions. If the authors discuss the C-terminus of EB1, they should analyze the whole C-terminus of EB1, which would strengthen the conclusion they have made.

Honapa et al., Cell, 2009, reported chemical shift perturbations (CSPs) on the peptide binding for the full EB1c fragment, which includes the negatively charged C-terminus. Similar to our study, they observed significant CSPs in the FVIP region but negligible CSPs at the negatively charged EEY end. They concluded that the final eight EB1c residues did not contribute to binding and used a truncated EB1c construct for their structural analysis. Building on that study, we used the same EEY-truncated construct to analyse the contribution of the C-terminus in more detail. We believe that conducting additional experiments with the full C-terminus with respect to SxIP binding would be superfluous, as it would merely replicate the findings of Honapa EA. We have added the rationale for selecting the truncated EB1c construct to the text, referencing Honapa et al.

**Reviewer #2 (Recommendations For The Authors):**
(1) Figure 2C: The authors can analyze the 11MACF peptide as well, to provide more assurance to their argument. It would be easier to distinguish the sequences of "SKIP" and "FVIP" by changing their colors.

Our relaxation analysis (Fig. 2C) focuses on the dynamics of the unstructured C-terminal region in both the free and complex forms. Further relaxation analysis of the peptide would not provide additional information on this, and would be complicated by the presence of free peptide in solution.

(2) Figure 3B: Acidic residues in EBH should be labeled.Page 6, line 11: If the authors insist that the acidic patch will influence the interactions between EB1 and the peptide, the data of the analysis using the entire EB1 C-terminus should be included, given that the C-terminal tail of EB1 is highly acidic.

To test the contribution of charge to binding, we conducted an ITC experiment at increasing salt concentrations. We observed a significant increase in Kd values when the concentration of NaCl increased from 50 to 150 mM, which supports our conclusion regarding the significant electrostatic contribution. This conclusion is independent of the presence or absence of the C-terminus.

As we explained earlier, Honapa et al., Cell 2009, conducted an NMR experiment on the full EB1c and observed no CPSs in the EEY region, indicating a negligible contribution from the EEY region to SxIP binding. Therefore, we think that additional experiments involving the entire C-terminus are unnecessary, as they would simply replicate the results of Honapa et al. We have added the rationale for selecting the truncated EB1c to the text, referencing Honapa et al.

It would be very difficult to label the acidic residues without enlarging 3B considerably. However, we do not think this is necessary as we are not discussing any specific residues. The current figure shows the distribution of the surface charge, which is sufficient for our purposes.

(3) Figure 2B (Page 4, line 27): The side chain of S5477 should be drawn. The authors should include a figure of the crystal structure of EBH and SxIP as a comparison (Honnappa et al., Cell, 2009). In their paper, Honnappa et al. performed chemical shift perturbation titrations by NMR. From their analysis, I imagine that the EB1 tail may not be critical for the EB1 C-terminus:SxIP interactions, since the signals in the tail are not significantly perturbed. The authors should cite this paper.

We are grateful to the reviewer for highlighting this. CSP analysis of the Honapa EA revealed significant changes in the FVIP region, which we also observed. They also reported negligible CSPs at the EEY end, demonstrating that this part of the tail is non-critical and can be removed. We have added text to the manuscript to highlight the similarity between CSPs and those observed in Honapa EA. Figure 2B shows the side chains for the residues with the strongest detected contacts. These do not include S5477.

(4) Figure 3C (ITC data): The stoichiometric ratios in the ITC data look strange. EBH vs KPSKIPVLLRKRK, is it 1:1?

We repeated the ITC experiments using a new stock of the peptide and a new batch of the protein, checking the concentrations using UV spectroscopy. The new experiments produced a stoichiometry close to 1, as shown in the table.

(5) Page 10, line 27: "The TPQ sequence of 11MACF is not optimal...": What is the meaning of "optimal"? The transient interaction between EB1 and its binding partner is responsible for the dynamics of the microtubule cytoskeleton. In a sense, the relatively weak interaction is "optimal" for the system. The authors should rephrase the word.

We agree that weak interactions are optimal from a functional perspective, as they have been selected through evolution. In our case, 'optimal' refers to the hydrophobic interaction with the C-terminus. We replaced 'optimal' with 'ideal' to draw more attention to the second part of the sentence, which clarifies the context.

(6) Page 11, line 2: "small number of comets enriched in the peptide that were too faint for the quantitative analysis, comparable to the reported previously (Honnappa, Gouveia et al. 2009)." Honnappa et al. used EGFP-fusion constructs in their study: EGFP forms a weak dimer, which presumably gave different results from the authors' mTFP-constructs. The authors can note this point in the text.

We are grateful to the reviewer for highlighting this. This aligns well with our conclusion that dimerisation is important for localisation to comets. We have added this point to the text.

(7) Page 10, line 21: The authors calculate the free energy of complex formation between EBH and MACF peptide and explain in the text, but it is hard to follow.

We simplified and clarified the description of the energy contributions by focusing on the SxIP and non-SxIP regions of the peptide, as well as the EBH C-terminus.

Minor points:Page 2, line 9: IP motifs are not usually located in the C-terminus. For example, SxIP in Tastin is located in the N-terminal region, and SxIPs in CLASP are in the middle.

We corrected this statement, removing C-terminal.

Page 3, line 4: The authors should note the residue numbers of SKIP.

We think that in this context the residue number of the SxIP region are not important and would be distracting.

Figure 3D and Figure S3F: Make the colors and the order the same between the two figures.

We changed the colour scheme and the order of ITC parameters in S3F to match the main figure.

Figure 1A, 2B, Figure S5: Change the color of SKIP from other residues in the same chain, otherwise the readers cannot distinguish. Likewise, change the color of FVIP in Figure 2B.

We think that changing the colours will complicate the figures unnecessary. The corresponding residues are clearly labelled in the figures.

Figure 3, Figure S5, S6, S7: Box the letters of SKIP for clarity.

We boxed the SxIP region in S5 (new S6) and underlined in S6 (new S7). In S7 (new S8) the location of SxIP is very clear from the homology.

Figure 3B; Figure S2: Hard to recognize the peptide (MACF in green).

We increased the size of 3D and S2, making it easier to see the peptide.

Figure 1C and D: Make the residual numbers of the x-axes the same between the two graphs.

We made new plots with a linear scale for the residue numbers.

Figure 2A: The structures shown are not EB1. It should be described as EBH or EB1(191-260 a.a.).

Corrected.

Page 5, line 17: "the S2 values of the C-terminus" should be "the S2 values of the C-terminal loop in EBH", otherwise it is confusing.

Corrected.

Page 6, line 27; Figure S3C and S6: Please indicate the assignments of the resonances from "253FVI255" in the Figures.

We labelled the peaks corresponding to the 253FVI255 region in figure S6 (new S7). Figure S3 shows EBH-ΔC that does not include this region.

Page 7, line 25: Figure S7 should be S8.

Corrected

Page 12, line 6: "sulfatrahsferases" must by a typo.

Corrected.